# Melanocytes and photosensory organs share a common ancestry that illuminates the origins of the neural crest
Yuliia Fatieieva[1], Rozalina Galimullina[1], Sergey Isaev[1], Alexander Klimovich [2], Laurence A. Lemaire [3] & Igor Adameyko [1,4] ✉

In vertebrates, two major cell types produce extensive pigmentation: neuroepithelium-derived retinal pigment epithelium (RPE) of the eye and neural crest-derived melanocytes. Both produce melanin, express opsins, and exhibit photosensory functions. However, the evolutionary relationship between these cells - whether pigmentation was coopted or they share a common ancestry - remains unclear. We explore these scenarios including the hypothesis of a shared origin from an ancestral pigmented photosensory structure. For this, we harness single cell transcriptomics, chromatin accessibility and spatial transcriptomics data, to connect the transcriptional programs in melanocytes, pinealocytes and RPE with that of the pigmented cells in the sensory vesicle of the tunicate *Ciona*. The results reveal common regulatory gene expression modules spanning beyond pigment production, including photoreception, metabolism and biosynthesis. This evidence does not favor a model where pigmentation was coopted into one of these cell types, and rather supports the homology of melanocytes and RPE. Further, phylotranscriptomics approach expose recently-evolved melanocyte-specific and RPE-specific functions, which diversified after these types split from the ancestral cell type. Overall, our results support that melanocytes and RPE evolved from ancestral pigmented photosensory structures in chordates, initiating the origin of the neural crest – a major evolutionary driver of the vertebrate lineage.

In chordates, the photosensory structures originate from neuroectoderm and require phototransduction and dark pigment (melanin), which provides shading of the photoreceptors and helps to adapt their dynamic range. Melanin and photoreception have been closely linked since the emergence of cnidarians, as both are present in cnidarian sensory organs, including the camera eyes of cubozoan jellyfish[1,2]. Therefore, the photoreception and melanin-based pigmentation programs are ancient, intertwined and are unambiguously associated with visual sense[3,4].

This is supported by the fact that melanin-containing cells are parts of the visual sensory organs in cephalochordates[5] and tunicates[6]. Often, photosensory cells themselves contain melanin pigment[6], which might tune the dynamic range of their photosensitivity and protect against oxidative stress. Even in vertebrates, the third eye, also known as a pineal gland, contains melanin-bearing pinealocytes[7] expressing pinopsins providing photosensitivity[8]. Given that pinealocytes are thought to evolve

from photoreceptors[9,10], the presence of shading pigment melanin turned out to be preserved in evolution of the third eye.

In modern camera type eyes, retinal pigmented epithelium (RPE) shades the retina and provides the directionality of photoreception[11]. Remarkably, there are two noncanonical opsins (RRH and RGR), which RPE expresses in addition to the melanin gene expression module[12]. This suggests that RPE originated from the ancient pigmented photosensory cells and later took part in a retinol conversion cycle in addition to the major role in pigment-based shading in camera type eyes (such shading prevents the retina from being exposed to a light from a back-side of each eye). Also, there are strong developmental connections between pigmented RPE and photoreceptors of a vertebrate eye, because both cell types develop from lateral neuroectoderm of the developing brain[11]. Furthermore, in salamanders, RPE can dedifferentiate after trauma and give rise to regenerated retina, including photosensory cells[13]. Taken together, in vertebrates, these cell

[1]Department of Neuroimmunology, Center for Brain Research, Medical University Vienna, 1090 Vienna, Austria. [2]Zoological Institute, Christian-Albrechts University of Kiel, 24118 Kiel, Germany. [3]Department of Biology, Saint Louis University, MO, 63108 St. Louis, USA. [4]Department of Physiology and Pharmacology, Karolinska Institutet, 17177 Stockholm, Sweden. ✉e-mail: igor.adameyko@meduniwien.ac.at

types appear to share a common origin during evolution, embryogenesis, and even regeneration.

However, there is a vertebrate-specific cell type that is seemingly outside of the photosensory system and yet produces large amounts of melanin – the melanocyte. Indeed, melanocytes are often wrongly considered to be cells providing only pigment with no immediate response to a visible light. This limited view is not supported by the current state of the field, as we now know that melanocytes express melanopsins, components of the phototransduction cascade and can quickly respond to a visible light by redistributing the pigment granules – thus providing a dynamic, light-responsive camouflage to a range of aquatic animals[1,4,14]. This mechanism of fast camouflaging is dissimilar to the mechanisms of tanning[14], which largely depend on a long-term production of melanin, its intercellular transport from melanocytes to keratinocytes, and activation of UVA/B stress response as one of the DNA damage[15]. Interestingly, melanocytes, similarly to classic photoreceptors, also harness the phototransduction cascade to respond to the visible light[3,16,17]. Based on these findings, melanocytes appear to represent a photosensory pigmented cell which became autonomous from supplying the CNS with visual information, and instead acquired additional roles in the body, such as camouflaging, sexual displays, and other behaviors.

Collectively, based on the capacity "to see" and respond to a visible light, and also because of the presence of a shading pigment, melanocytes might be homologous to ancient pigmented photoreceptors, quite similar to those found in extant tunicates (within sensory vesicle) and cephalochordates (Hesse ocelli)[6,18,19].

Despite their remarkable similarities, melanocytes, unlike RPE or pinealocytes, develop from vertebrate-specific neural crest cells – a unique population of multipotent migratory progenitors originating in the dorsal neural tube and undergoing epithelial-to-mesenchymal transition (EMT) to invade the peripheral tissues of the developing animals[20]. Neural crest cells literally shaped "vertebrates" as we know them, with jaws, teeth, faces, autonomous and somatosensory nervous systems[21]. The neural crest became one of the most important innovations and the 4th germ layer in the lineage of chordates[22]. Resolving the evolutionary origin of the neural crest is a task of the utmost importance, and it might be addressed by understanding the evolution of melanocytes.

Because the embryological origin of RPE and pinealocytes is different to a significant extent from the neural crest and neural crest-derived melanocytes, such disconnection in terms of developmental origin is a major factor which precludes homologizing RPE cells, pinealocytes and scattered melanocytes. However, is there such an unbridgeable divide, and is it worth revisiting the homology of these cell types in light of the most recent knowledge?

During the course of development, the neuroepithelium generates retina and RPE[23]. Similarly, melanocyte-producing neural crest cells arise from the neuroepithelium at the dorso-lateral neural tube border[24,25]. In mammalian embryos, neural crest cells migrate after neural tube closure in the posterior region but prior to closure in the anterior region[26], allowing the roof plate to form at the dorsal midline once the neural crest cells have exited the developing tube[27]. It is reminiscent of the development of the pineal gland, which is positioned in an unpaired fashion at the level of the dorsal midline[28–30] – near where the roof plate typically forms and the neural crest cells emigrate.

Apparently, both neural crest cells, which give rise to melanocytes, and pigmented photosensory organs originate from embryonic neuroectodermal tissue, although they emerge from distinct, differentially patterned sub-regions. However, neural crest cells (along with their intermediate multipotent derivatives, such as Schwann cell precursors) have evolved as an intermediary developmental stage between the embryonic neuroepithelium and the terminal photosensory pigmented melanocytes. This intermediate step likely facilitates the widespread dispersal of melanogenic progenitors throughout the body, including the skin. Such photosensory pigment cell dispersal could be potentially driven by unanticipated photoreception and pigmentation requirements that early chordates faced[3], which eventually led to the elaboration of multipotent neural crest as a 4th germ layer. Indeed,

such reasoning might eventually shed some light on the mystery of the general evolutionary origin of the neural crest in vertebrates.

Here we suggest that changes in visual sense in combination with advantages of dynamic camouflaging could be the evolutionary force that partly contributed to the origin of melanogenic neural crest in early vertebrates, which led to scattered melanocytes instead of the CNS-residing pigmented ocelli. Based on this hypothesis, it is possible to test the homology of vertebrate RPE and melanocytes at the level of transcriptional regulation. To address this, we took advantage of published single cell transcriptomics and spatial transcriptomics datasets of RPE and melanocytes, and of the sensory vesicle from ascidian *Ciona intestinalis*. We further applied techniques of comparative single cell transcriptomics and phylostratigraphy analysis to test the homology between melanocytes and photosensory structures, for which we found supporting evidence.

## Results

Melanocytes and RPE cells share key functions such as photoreception and pigmentation[3], which hint at a common evolutionary origin at the dawn of vertebrates. According to a current tradition, homology refers to the identity of structures in different species due to common ancestry, which may not be immediately apparent in distantly related species, where structures may have become dissimilar due to evolutionary divergence[31,32]. This is possibly the case for RPE and melanocytes, which share conserved functional traits, yet are vastly different in morphology, organization (epithelial vs mesenchymal) and development. Investigation of homology of distantly related cell types requires a detailed comparison of their transcriptional programs and modules.

To compare the transcriptional programs in RPE cells and melanocytes, we conducted a re-analysis of a published single cell transcriptomics dataset by Lee et al., which includes both cell types from the same batches of adult murine anterior eye chamber (Fig. 1a)[33]. The eye is a strategic location for this study because it contains a high density of neural crest-derived melanocytes near the retina and RPE, making it an ideal system for examining the distinct and overlapping gene expression programs of these cells[34]. We would like to emphasize that analysis of both RPE from the eye and nearby melanocytes from the same sample advantageously eliminates technical batch effects.

The re-analysis involved clustering the cells from the dataset, revealing the presence of several expected cell types, including fibroblasts (mesenchymal cluster), vascular and perivascular cells, Schwann cells, immune cells, and importantly, melanocytes and RPE cells (Fig. 1b–d, Supplementary Fig. 1). The clusters identified as RPE and melanocytes were characterized by the expression of well-established marker genes. For RPE, classic markers such as *Rpe65* and *Otx2* were predominant, whereas melanocytes exhibited stronger expression of *Tyr, Mlana, Dct,* and *Pmel*, which are essential for melanin biosynthesis and function (Fig. 1c–f).

Using STRING analysis to evaluate the functional links between differentially expressed genes in RPE and melanocyte clusters, we found that melanocytes displayed an overrepresentation of an interconnected gene set essential for melanin synthesis and distribution. Conversely, RPE cells were enriched in networks related to visual perception and associated metabolic pathways (Fig. 1g). Gene ontology enrichment analysis supported these findings, showing that the melanocyte cluster was specifically enriched in processes related to melanin synthesis and melanocyte differentiation (Fig. 1h). In contrast, the RPE cluster was enriched in processes related to retinol metabolism and visual perception (Fig. 1i). This result aligns with the established roles of RPE cells in the retinoid cycle, which is essential for photoreception in the eye. This cycle is responsible for regenerating 11-cis-retinal, the chromophore that binds to opsins in photoreceptor cells, allowing these cells to detect light. The RPE serves multiple functions in this cycle, including retinoid metabolism, storage, and recycling[35]. Importantly, RPE cells also expressed the melanin synthesis program, though at lower level as compared to melanocytes (Supplementary Fig. 1).

Further gene expression analysis identified a subset of genes commonly expressed in both melanocytes and RPE cells (Fig. 2a). This is relevant for testing whether cooption of a specific gene circuit occurred, or whether

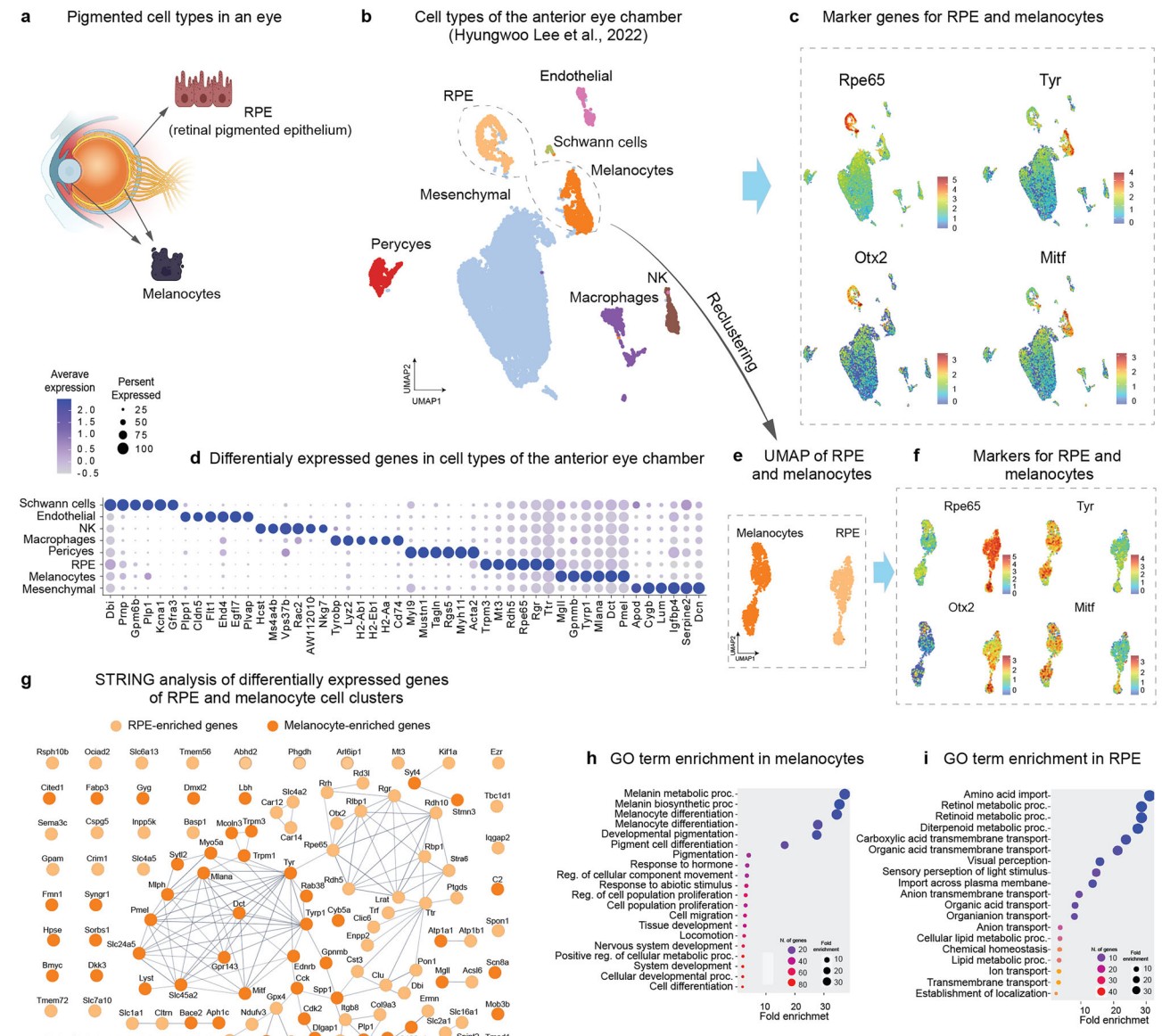

**Fig. 1 | Single cell analysis of RPE and melanocytes from the murine eye.**
**a** Schematic representation of a vertebrate eye with uveal melanocytes and RPE layer. Created with BioRender.com. Licensed content used with permission. **b** Clustering of single-cell transcriptomic data adapted from Lee et al., 2022. **c** UMAP visualizations of main melanocyte and RPE markers. **d** Dot plot of major marker genes for clusters in **b**. **e**, **f** UMAP (**e**) after re-clustering of melanocyte and RPE clusters from Lee et al. and differential expression of key genes (**f**). **g** STRING analysis of differentially expressed genes enriched in melanocytes or RPE cells (the color code links genes to cell types). **h**, **i** GO terms enrichment of genes expressed in melanocytes (**h**) and RPE (**i**).

diverse classes of gene functions were shared, which would instead support homology (Fig. 2b). While these genes may exhibit stronger enrichment in either melanocytes or RPE, they are distinctly expressed in both cell types when compared to other cellular populations of the anterior eye chamber. Gene Ontology and STRING analysis of the common melanocyte (MEL) and RPE genes revealed that melanocytes and RPE shared metabolic, molecular transport, matrix interaction, photoreception and other signal transduction modules beyond a shared melanin synthesis program (Fig. 2c, d), as well as a basic set of transcription factors, including *Mitf, Sox10, Smarca4, Bhlhe41, Eno1*, and *Got1* (Fig. 2e). A half of these shared transcription factors is directly related to melanin biosynthesis control and distribution (*Mitf, Sox10, Smarca4*), whereas the remaining factors (*Bhlhe41, Eno1, Got1*) are involved in other cellular functions not directly connected to melanin (Fig. 2f).

Overall, our analysis revealed that melanocytes and RPE share not only a genetic program related to melanin-based pigmentation but also

numerous other programs associated with a wide array of cellular functions. Notably, the proportion of pigmentation-related genes among all differentially expressed genes shared by RPE and melanocytes was only 10.5%, indicating that these cell types share a broader functional repertoire beyond pigmentation. This result rather supports that RPE and melanocytes might share the evolutionary origin, and argue against an alternative - a simple co-option of a pigmentation gene expression circuit into some enigmatic cell type in basal chordates.

To gain a more comprehensive view of the gene regulatory logic in these cell types, we performed comparative analysis of GRNs from melanocytes, RPE and other cell types from the adult murine anterior eye chamber. For this, we again re-analyzed the published single cell transcriptomics dataset by Lee et al[33]., this time embedding the transcriptomes based on the list of all DNA-binding proteins (mainly transcription factors and chromatin remodeling proteins) (Fig. 3a). The clustering based on this manually selected subset of genes (DNA-

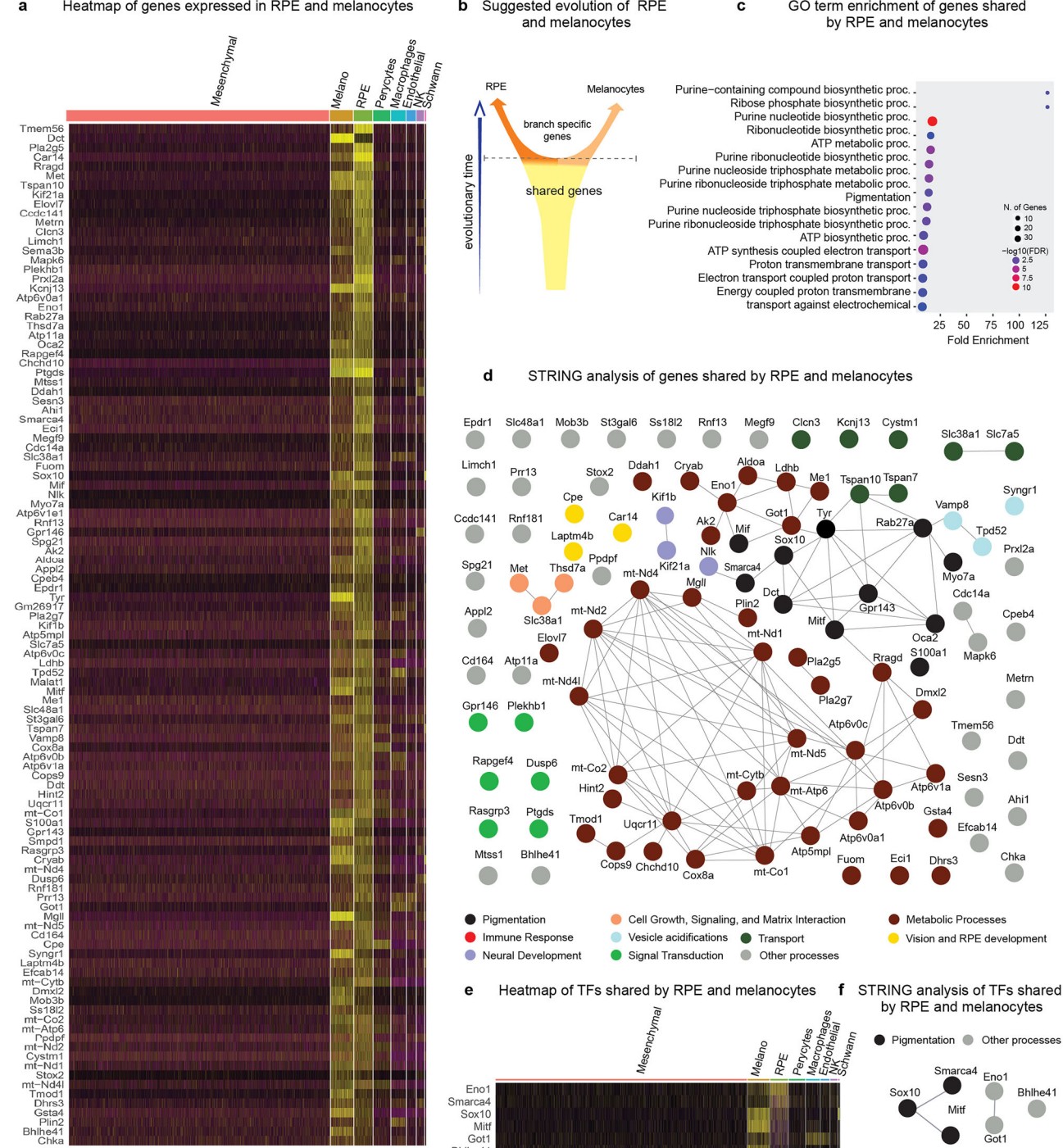

**Fig. 2 | Analysis of differentially expressed genes that are shared between melanocytes and RPE cells. a** Gene expression heatmap showing genes, which are specifically and jointly expressed in both melanocytes and RPE clusters. **b** Scheme of a hypothesis: these shared genes may reflect homology and a common origin of melanocytes and RPE. **c** GO terms enrichment of specific genes expressed by both melanocytes and RPE. **d** STRING analysis of these shared genes. **e, f** Heatmap (**e**) and STRING analysis (**f**) of expressed transcription factors specifically shared by RPE and melanocytes. Heatmaps reflect log-normalized gene expression values.

binding) recovered all previously resolved cell types (based on all genes, as shown in Fig. 1b). In this case, the structure of hierarchical clustering showed that RPE, melanocytes and Schwann cells shared the neighboring positions in a graph, with Schwann cells and melanocytes being the closest neighbors (Fig. 3b). We do not find this result surprising because melanocytes are derived from Schwann cell precursors during embryonic development (and they both belong to the neural crest lineage)[36,37]. However, the hierarchical clustering based on single cell differential gene expression analysis is not a perfect tool to infer the evolutionary and

developmental relationships between cell types based on the sets of expressed transcription factors. Indeed, despite its intuitive appeal, there are several downsides and limitations associated with hierarchical clustering approach when used in the context of defining evolutionary relationships between cell types. Hierarchical clustering can be highly sensitive to transcriptional noise and fine differences in gene expression levels. Additionally, the outcome of hierarchical clustering is heavily dependent on the choice of distance metric and linkage method[38]. This dependence introduces subjectivity and potential bias in defining

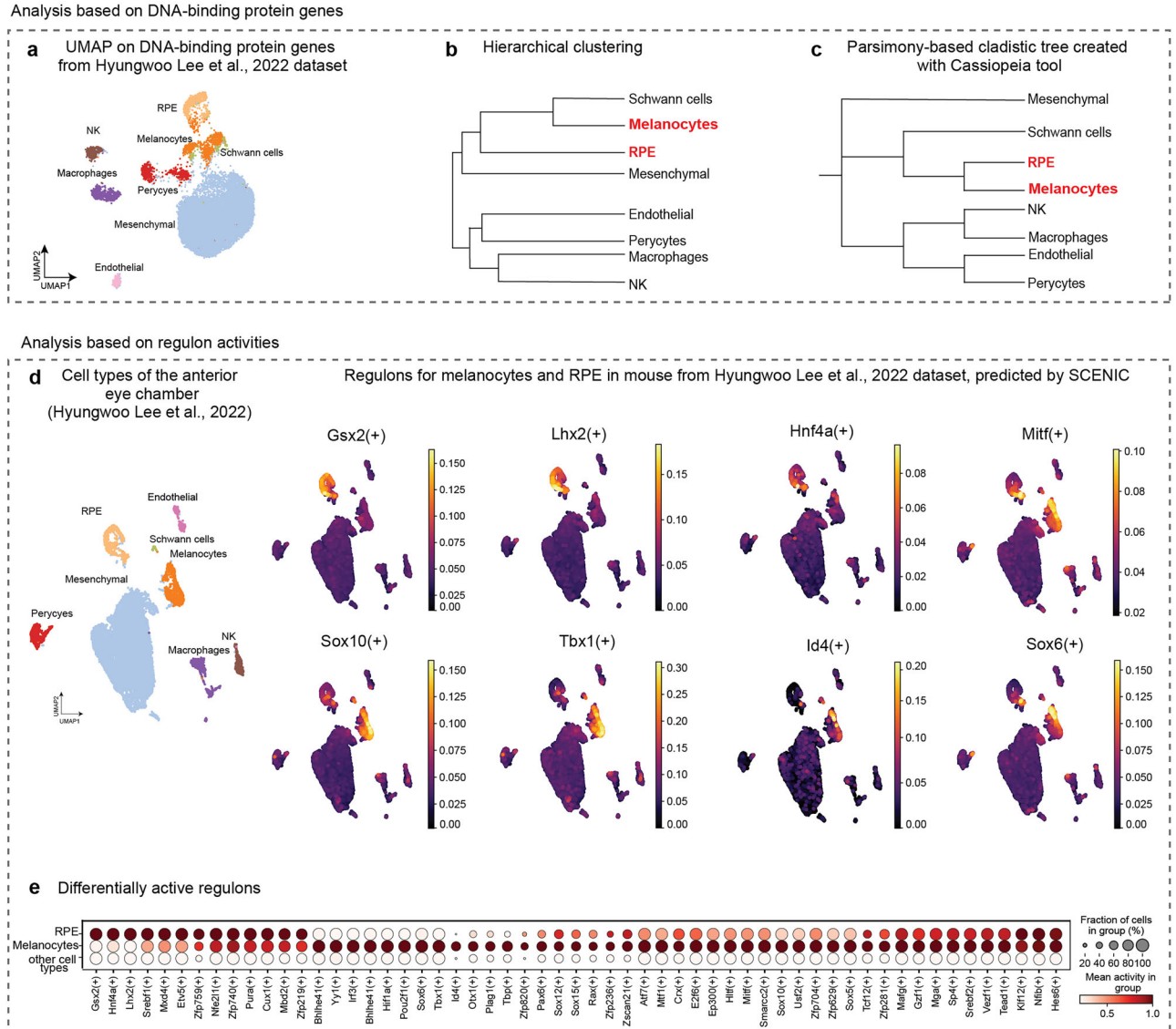

**Fig. 3 | Clustering of eye single cell data from Lee et al., 2022 using only DNA-binding protein-coding genes (mostly transcription factors). a** UMAP based on DNA-binding protein genes. **b** Hierarchical clustering diagram showing similarity between clusters representing major cell types including melanocytes and RPE. **c** Parsimony-based cladistic tree of cell types built using transcription factor code information and CASSIOPEIA tool. **d** Clustering of single-cell transcriptomic data from Lee et al., 2022 and subsequent SCENIC analysis of transcription factor activity (regulons) in melanocytes and RPE cells. **e** Differentially active regulons in RPE, melanocytes and other cell types in the anterior eye chamber represented as a dot plot.

evolutionary distances, as different metrics and methods might not align with biological reality.

Other phylogenetic tree reconstruction methods utilizing cladistics analysis are based on parsimony criterion, which appears to be a better alternative[39]. Building a maximum parsimony tree means finding the tree that requires the fewest evolutionary changes to explain the observed data[40]. Using parsimony, we can directly compare the sets of expressed transcription factors, where every TF is either present or absent[41]. We generated such sets for every cluster by filtering the single cell data applying stringent criteria (see Methods section for details). Next, we performed the analysis using a parsimony module in the CASSIOPEIA tool[42]. CASSIOPEIA is a computational framework designed to reconstruct phylogenetic trees from single-cell transcriptomics data, particularly leveraging the information encoded in CRISPR-based lineage tracing systems. The parsimony module within CASSIOPEIA is one of the core components aimed at reconstructing the most parsimonious tree structure, which means finding the tree that requires the fewest evolutionary changes to explain the observed data[40]. The resulting tree revealed the potentially shared evolutionary ancestry of

melanocytes and RPE cells based on the presence/absence of transcriptional factors (GRN composition), and these cell types appeared as closest neighbors on a generated tree (Fig. 3c). At the same time, the Schwann cells emerged as the most proximal outgroup.

We next sought to determine whether the observed transcriptional similarity between retinal RPE cells and ocular melanocytes reflects a shared evolutionary origin, or alternatively, results from convergent transcriptional programs driven by a common tissue environment - specifically, their co-localization within the eye and exposure to similar signaling niches. To address this question more rigorously, we extended our comparison beyond ocular tissues by incorporating skin-derived melanocytes into the analysis. If RPE cells cluster more closely with melanocytes from distant anatomical sites, such as the skin, this would support the notion of deep lineage conservation, rather than local convergence due to tissue-specific niche signals.

To systematically compare RPE cells with skin melanocytes in the context of diverse cellular lineages, we merged a dataset from the anterior eye chamber containing RPE cells (GSM5560840, GEO / (Lee et al., 2022)[33]) with four distinct clusters of skin-derived melanocytes obtained from

published sources. These included: melanocytes from back skin at postnatal day 0 (GSE131498, GEO / (W. Ge et al., 2020)[43]), postnatal day 4 back skin melanocytes (GSE181390, GEO / (Lee et al., 2024)[44]), activated melanocyte stem cells from anagen II stage hair follicles in 5-week-old mice (GSE203051, GEO/(Sun et al., 2023)[45]), and quiescent melanocyte stem cells profiled via deep Smart-seq2 sequencing (GSE147298/(Infarinato et al., 2020)[46]). Each melanocyte cluster from these studies was integrated into the analysis independently (Supplementary Fig. 2).

Hierarchical clustering of the merged data consistently positioned RPE cells in proximity to both ocular and skin-derived melanocytes, relative to all other cell types. After constructing a dendrogram based on hierarchical clustering of global transcriptional profiles (considering all genes), we observed that RPE cells, melanocytes from both ocular and skin origins, and Schwann cells consistently clustered in close proximity (Supplementary Fig. 2a–c, e–g, i–k, m–o). The close association of Schwann cells is biologically anticipated, as melanocytes are known to arise developmentally from the Schwann cell lineage[36,47]. Such persistence of transcriptional similarity reflects their immediate shared developmental history, which lends additional confidence to the robustness and biological relevance of our analysis, demonstrating that developmentally defined relationships can be faithfully recovered through unbiased transcriptome-wide comparisons.

Importantly, maximum parsimony-based analysis - focusing specifically on transcription factor profiles - consistently grouped RPE cells with skin-derived melanocytes across all tested conditions (Supplementary Fig. 2d, h, l, p). In this analysis, skin melanocytes often exhibited greater transcriptional similarity to RPE cells than did ocular melanocytes. This result indicates that melanocytes from anatomically distant sites retain a transcriptional program or signature that aligns them closely with RPE cells despite occupying distinct tissue environments.

To explore the commonalities of gene expression regulation in RPE and melanocytes, we performed an analysis of regulons. A regulon includes a transcription factor and the genes it directly regulates[48]. These target genes often share common regulatory motifs in their promoters or enhancers bound by the transcription factor. SCENIC (Single-Cell Regulatory Network Inference and Clustering) is a computational tool designed to infer regulons and gene regulatory networks from single-cell RNA sequencing data. SCENIC identifies candidate upstream transcription factors (TF) for each gene in the network using known TF-binding motifs. This leverages databases of DNA motifs and transcription factor binding sites, as well as ChIP-seq data[48].

After calculating the full set of active regulons emerging in the Lee et al. dataset (by using SCENIC tool), we selected regulons that were active in melanocytes, or RPE, or in both, but not in other unrelated cell types. Surprisingly, the vast majority of regulons were shared between melanocytes and RPE cells, with only few that were either melanocyte- (Id4 + , Tbx1 + , Sox6+ and others) or RPE-specific (Gsx2 + , Hnf4a + , Lhx2 + ) (Fig. 3d-e). This result points towards a profound similarity in both gene expression profiles and upstream regulatory GRN landscapes[49] among melanocytes and RPE cells, extending beyond pigmentation and reinforcing their shared evolutionary origin.

Also, analysis of a developmental time course from previously published single cell and spatial transcriptomics data[50–52] showed that gene expression profiles from early stages of RPE development are similar to that of melanocytes. For instance, RPE from human developing eye at early weeks 4−8 showed reduced or missing expression of RPE-specific genes (Fig. 4a-b), which were eventually expressed later in human development—during weeks 12–21 (Fig. 4c-e). This observation is consistent with spatial transcriptomics results from mouse embryos at E14.5 and E16.5, showing that eye pigmentation (melanocyte)-related markers appeared before RPE-specific markers (Fig. 4f-g). This suggests that the function-related distinctions in RPE and melanocytes emerge upon reaching maturity, and that these cell types might share a developmental semi-convergent state at some point, despite generally different developmental trajectories.

This result inspired us to explore where else the joint photosensory and pigmentation machinery is active. For this we turned to the photosensory endocrine cells of the vertebrate third eye—the pineal gland, which was previously reported to show a variable degree of weak melanin-based pigmentation in different vertebrate species[8]. The re-analysis of published single cell transcriptomics datasets of an adult primate anterior eye chamber[33,53] together with co-clustered pineal gland single cell data showed that pinealocytes, while being endocrine and photosensory, also share a portion of the melanocyte and RPE gene expression program, including transcriptional control of pigmentation (Mitf and Sox10) and a set of pigmentation production-related genes (Dct, Tyr, Pmel) (Supplementary Fig. 3a, b). Importantly, the other non-pigmentation-related shared RPE/ melanocyte genes were also expressed in pinealocytes, such as Padi2, Meis2, Dlgap1 (Supplementary Fig. 3c). This result strengthens the evolutionary connection between melanocytes and other photosensory pigmented cell types, including RPE and pinealocytes.

Changes in the sequences of distal regulatory elements have been recognized as major drivers of evolutionary diversification across species[54]. Thus, we explored the regulatory landscape of RPE and melanocytes in a comparative context, focusing on chromatin accessibility and non-coding regulatory elements that may contribute to their transcriptional similarity and evolutionary relationships. To achieve this, we utilized the recently published high-resolution 10x Multiome dataset from Mullin et al. 2023[55], which simultaneously profiles chromatin accessibility (ATAC-seq) and gene expression (single-nucleus RNA-seq) from the mouse eye (Supplementary Fig. 4). This dataset offers comprehensive coverage of ocular melanocytes, RPE cells, and a broad range of neuroretinal and anterior eye chamber cell types. A key advantage of this multiomic approach is the ability to confidently assign cell identities based on transcriptomic signatures while analyzing chromatin accessibility in the same nuclei, thus overcoming limitations inherent to ATAC-seq-only data, such as low cell-type resolution and susceptibility to misclassification.

Our initial dimensionality reduction analysis, based solely on ATAC-seq embeddings, revealed substantial misattribution between RPE and melanocyte clusters, with cells frequently grouped into one another's domains (Supplementary Fig. 4a-b). However, by cross-referencing transcriptomic profiles from the same nuclei, we were able to resolve true cell identities with higher confidence. These observations prompted the hypothesis that RPE cells and melanocytes share a significant fraction of accessible chromatin regions, particularly in non-promoter, non-coding elements, which may underlie their apparent proximity in ATAC-based embeddings. To formally assess this, we constructed a dendrogram using a maximum distance tree built from chromatin accessibility data restricted to non-promoter intergenic regions, because promoter-related regions will reflect overall expression level of corresponding genes. This analysis revealed that RPE cells, melanocytes, and Schwann cells clustered on the dendrogram in close proximity (Supplementary Fig. 4c). The focused investigation of open chromatin regions in the upstream and downstream regions of Sox10—a transcription factor with a key role in both melanocytes and RPE cells – revealed mostly common and some distinctly open chromatin regions linked to promoter accessibility in peaks-linked-to-genes analysis (Supplementary Fig. 4d).

To further explore the robustness of the cell type dendrogram based on chromatin accessibility profiles, we applied an alternative strategy based on Euclidean distances, using the same subset of accessible chromatin regions. Unlike the maximum distance tree, this method unexpectedly placed RPE cells as a highly distant outgroup, producing a dendrogram that was less biologically interpretable with respect to RPE relationships (Supplementary Fig. 4e). This discrepancy likely reflects the known limitations of ATAC-seq data—particularly its sparsity and sensitivity to technical noise—highlighting the importance of integrative approaches for evolutionary interpretation. Also, the evolution of genomic regulatory sequences is expected to be far less stable over longer periods of animal diversification as compared to cell identity codes driven by transcription factor networks, despite some degree of conservation of the overall regulatory landscape in homologous cell types.

To examine the functional relevance of the accessible chromatin regions shared between RPE cells and melanocytes, we performed gene ontology (GO) enrichment and STRING analysis using genes proximal to

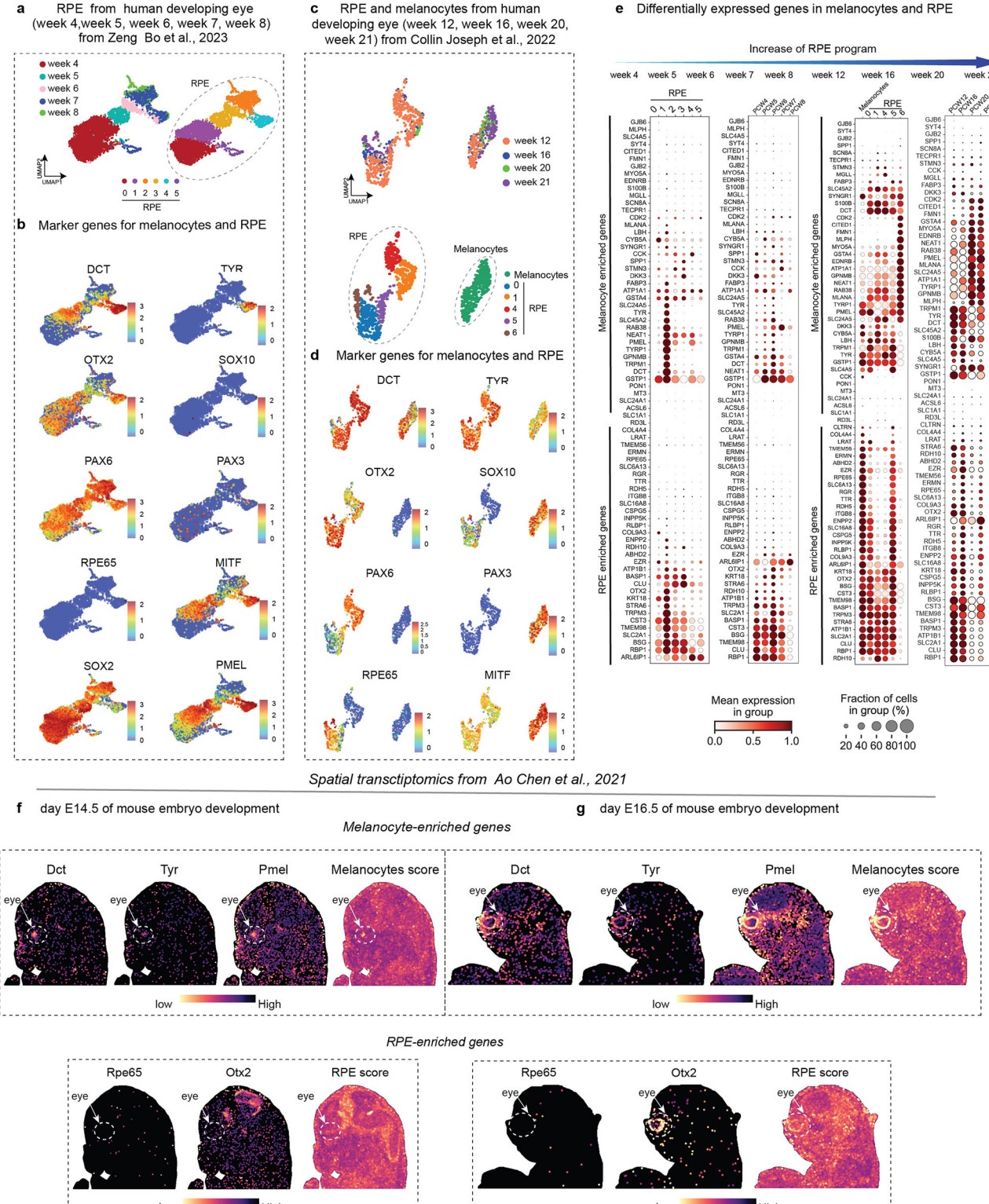

**Fig. 4 | The developmental timeline of RPE in comparison with ocular melano-cyte gene expression programs. a** UMAP and clustering of early developing human RPE from Zheng et al., 2023. **b** Expression of the main RPE and melanocyte marker genes from the same dataset (as in **a**). **c, d** UMAP and clustering of later developmental stages of human RPE and melanocytes from Collin et al., 2022. **e** Dot plot showing expression of RPE-enriched and melanocyte-enriched genes across clusters and developmental stages (PCW - post conception week). **f, g** Analysis of melanocyte- and RPE-enriched genes in mouse spatial transcriptomics data from Chen et al., 2021.

open chromatin regions (Supplementary Fig. 5). These revealed significant enrichment of pathways related to pigmentation, cytoskeletal organization, and immune system signaling—biologically plausible categories that align with the known properties of both cell types.

In contrast to less stable ATAC-seq-based dendrograms, transcriptomic-based comparisons including those using parsimony-based criterion produced more stable and consistent clustering patterns. In the previous analysis, RPE cells grouped reliably with both ocular and

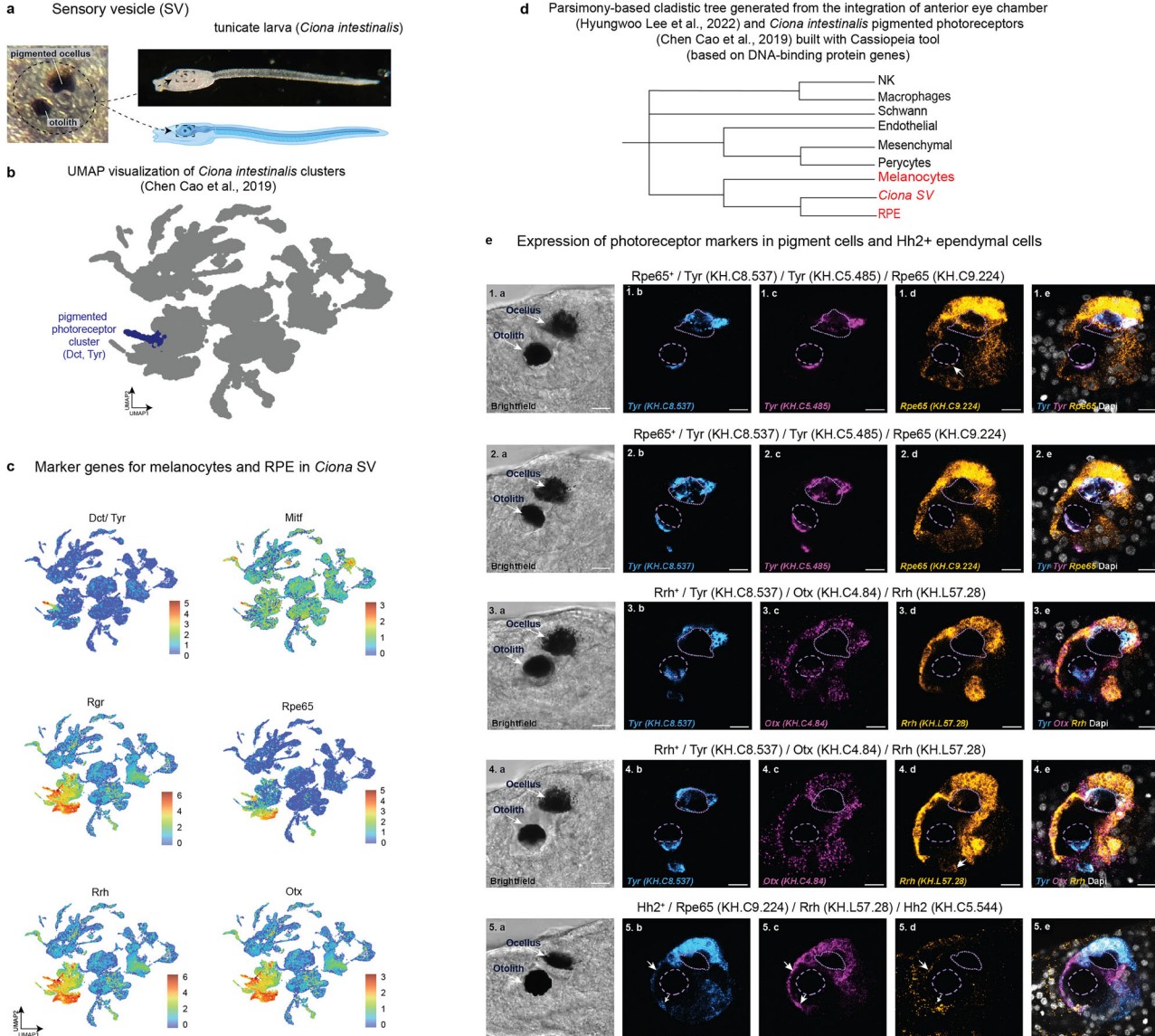

**Fig. 5 | Comparative evolutionary analysis of photosensory pigmented cells from the sensory vesicle of ascidian tadpole larva based on single-cell transcriptomics data by Cao et al., 2019. a** The position and structure of the sensory vesicle (SV) in the *Ciona intestinalis* tadpole larva. Created with BioRender.com. Licensed content used with permission. The only melanized structures in tunicate larvae are pigmented ocellus and otolith. **b** UMAP of Chen Cao et al., 2019 dataset and identification of the cluster corresponding to pigmented photoreceptors based on expression of the paralogs of *Dct, Tyr, Mitf, Otx, Rrh.* **c** UMAPs showing the expression of melanocyte- and RPE-associated marker genes in *Ciona* pigmented photoreceptors. **d** Cladistic tree of cell types obtained via parsimony-based analysis of sets of expressed transcription factors using the CASSIOPEIA tool. **e** Expression of photoreceptor-related genes visualized by HCR in the SV of *Ciona* larvae. Confocal optical sections show mRNA expression of two *Tyr* paralogs (identifier *KH.C8.537*, 1.b and 2.b, blue; identifier *KH.C5.485*, 1.c and 2.c, magenta), and *Rpe65*

(1.d and 2.d, yellow). Larva in 1.a-1.e expresses *Rpe65* in ocellus and otolith (white arrow, $n = 9/37$ larvae). Larva in 2.a-2.e expresses *Rpe65* in the ocellus, not the otolith (28/37 larvae). 3.a-4.a: Optical section of representative larval sensory vesicles stained for one *Tyr* paralog (3.b and 4.b, blue), *Otx* (magenta), and *Rrh* (yellow) by HCR. *Otx* is present in both cells ($n = 33$ larvae). The ocellus and not the otolith in 3.a to 3.e expresses *Rrh* ($n = 31/33$ larvae), while both pigment cells in 4.a to 4.e express *Rrh* ($n = 2/33$ larvae). The white arrow in 4.d points to the *Rrh* expression in the otolith. 5.a-5.e: Optical section of the sensory vesicle of a representative larva stained for *Rpe65* (5.b, blue), *Rrh* (5.c, magenta), and *Hh2* (5.d, orange) by HCR. White arrows point to *Hh2+* ependymal cells expressing *Hh2, Rpe65, and Rrh* ($n = 28$ larvae). Panels 1.a, 2.a, 3.a, 4.a, and 5.a are brightfield images. Panels 1.e, 2.e, 3.e, 4.e, and 5.e are merged images of the fluorescent channels combined with the nuclear dye Dapi (white). Dashed lines represent the pigment of the otolith and ocellus in the fluorescent image. Scale bar 10 μm.

skin-derived melanocytes across all tested conditions. To further assess such constructed tree's ability to capture known evolutionary relationships, we extended the dataset to include a broader range of cell types, such as rod and cone photoreceptors, retinal neuroglia, and immune cells from Li et al., 2024[56] (Supplementary Fig. 6a, b). After integrating retinal dataset by Li et al. 2024[56] with the anterior eye chamber dataset by Lee et al. 2022[33] from Fig.1 (Supplementary Fig. 6c), we successfully recapitulated previously proposed phylogenetic relationships, including the close pairing of rods and cones[57], the grouping of astrocytes with Müller

glia[58], and the segregation of various immune cell classes, providing additional support for the robustness of the method (Supplementary Fig. 6d).

Next, to expand the evolutionary framework, we selected the animal outgroup that would be evolutionary proximal to the split of vertebrate lineages from other chordates to investigate the transcriptional programs in primordial pigmented photoreceptors (Fig. 5a). The tunicates appeared to be a suitable group, for which the previously published single cell transcriptomics data from *Ciona intestinalis* include cells of the sensory vesicle,

the only pigmented structure in the *Ciona* larva[59]. The sensory vesicle is an integrated structure with various cell types, including pigmented photo-receptor, otolith and gravity sensors (Fig. 5a), working together to enable the tadpole to respond to its environment effectively. The re-analysis of this dataset revealed a well-defined cluster with pigmentation and photosensory programs, which we further used in our comparative analysis (Fig. 5b, c). For this, we identified orthologous genes between species and assigned the murine gene names to *Ciona* genes for convenience of downstream analysis. Because of impossibility to directly integrate the murine and *Ciona* single cell data or rely on label transfer procedures for evolutionary comparisons, we searched for other approaches. First, we found that the orthologs of *Dct, Tyr, Rrh, Otx, Rpe65, Mitf* and other melanocyte/RPE markers were expressed in pigmented ocellus cluster of *Ciona* (Fig. 5c). The GO terms enrichment and STRING analyses further confirmed the presence of expressed genes involved not only in pigmentation, but also in other pro-cesses shared by vertebrate melanocytes, RPE and *Ciona*'s pigmented photoreceptor lineage (Supplementary Fig. 7a, b). Finally, to investigate the shared regulation of transcriptional programs in murine melanocytes, RPE and pigmented ocellus of a tadpole larva of *Ciona intestinalis*, we opted for a parsimony analysis using a set of all transcription factors present in com-pared murine and tunicate cell types. This analysis revealed that melano-cytes, RPE and *Ciona*'s pigmented ocellus occupied the neighboring positions on the resulting dendrogram (Fig. 5d), supporting their homology.

In this part of the study, we used the larva of *Ciona* as an outgroup to vertebrates and focused on the co-expression of a selected set of marker genes that characterize melanocytes and RPE in vertebrates. Experimental validation is particularly important here because orthologous genes in basal chordates may differ spatiotemporally in their expression, and might never be physically co-expressed. To investigate this, we performed in situ hybridization chain reaction (HCR) using probes for several key transcripts known to be associated with pigmented and photosensory cell types in vertebrates. These included *Tyr* (involved in melanin biosynthesis), *Otx* (a marker of the anterior neural tube), *Rpe65* (a canonical marker of the RPE), *Rrh* (encoding a photoreceptive opsin), and *Hh2* (marking ependymal neural progenitors). The expression patterns of *Otx, Rpe65* and *Rrh* revealed a striking co-localization with *Tyr*, in the pigmented cells of *Ciona* larvae (Fig. 5e). Moreover, *Hh2* expression overlap with this domain, consistent with some of these ependymal neural progenitors sharing a common developmental lineage with the pigment cells[60]. These findings reinforce the central hypothesis of our study by supporting the potential homology between the pigmented ocellus of *Ciona* and the vertebrate RPE and mel-anocyte lineages.

In the next step of our analysis we implemented a phylostratigraphy approach to investigate the ancient and more recently-evolved components of gene expression programs to obtain the insights into diversification of functions in melanocytes and RPE cells. Phylostratigraphy is a computa-tional approach used to estimate the evolutionary age, or "birthdate," of genes and gene expression programs[61–65]. It involves mapping genes onto a phylogenetic tree to determine the evolutionary origin of each gene (attri-buting it to a specific evolutionary phylostratum), thereby enabling the study of gene emergence and the evolution of gene expression patterns across different lineages (Fig. 6a). Therefore, we wanted to explore which evolu-tionary events contributed to the individuation of melanocytes and RPE cell types from a joint ancestral program. Our hypothesis is that some evolu-tionary young genes playing a role in such later diversification of programs and functions may reveal the relevant functional modules. For addressing that, we took all melanocyte- or RPE-specific genes and plotted them according to their evolutionary age. As a cut-off line, we selected phylos-tratum 10, which demarcates a phylogenetic level at which vertebrate lineage emerged from chordate ancestors[65,66] (Fig. 6b). This time in evolution is expected to correspond to a lineage split into pigment-producing camou-flaging cells and RPE/pigmented photoreceptor cells of the main visual organs. Therefore, the genes which originated around phylostratum 10 and eventually incorporated into gene expression programs specific for either cell type, shall tell us about melanocyte or RPE-related functions evolving

after the evolutionary cell lineage split and therefore contributing to a cell type evolution.

The STRING tool and GO terms enrichment analysis showed that in the case of melanocytes, several genes were added to a pigmentation gene expression program at the dawn of vertebrates: those include *Mlana* and *Gpnmb* (Fig. 6c, d). The *Mlana* gene encodes a protein that plays an important role in the interactions with melanocyte protein *gp100* and is a key for stage II melanosome biogenesis[67]. GPNMB is a transmembrane glycoprotein that is important for melanosomes, and the knockdown of the corresponding mRNA inhibits the melanosome formation[68]. Also, there is evidence that GPNMB protein plays a role in adhesion of melanocytes to keratinocytes[69]. Overall, these results suggest that one of the recently evolved functions in melanocytes includes the formation of specific melanosomes capable of being transferred into keratinocytes, which is distinct from basic functions of pigmented photoreceptors, which need to keep melanin for the sake of their own shading. It seems that once melanocytes started to spe-cialize for camouflage and protection from UV, they evolved the exporting function, trading melanosomes to epidermal cell layers.

When it comes to recent functions that evolved in RPE cells around the 10th phylostratum, those include signaling and pro-neurogenic modules, as well as a response to injury, which corresponds to the well-established role of RPE in eye or even retinal regeneration in fish and amphibians (Fig. 6e, f).

In general, it seems that melanocytes and RPE cells in addition to diversifying the more ancient functions such as roles in vision (photo-reception), photoreceptor shading, or photoreception-specific metabolism (retinol conversion), melanocytes and RPE cells eventually evolved novel capacities linked to pigment export aiding dynamic camouflaging, more effective UV protection, or aiding regenerative potential of visual sensory organs.

Finally, we investigated the nature of all non-specifically expressed young genes (10+ phylostratum) incorporated into melanocyte and RPE transcriptomes. The phylostratigraphy analysis showed that the vast majority of these genes interface with the immune system, including cytokine-mediated immune response, antigen processing, and presentation, aiding the development of immune defense and histocompatibility in the vertebrate lineage (Fig. 6g–i). For instance, recent evolution of the vertebrate immune system is driven by diversification in MHC genes, as well as by the development of signaling interfaces between vertebrate immune system-specific cell types and tissue-resident partner cells[70,71]. Thus, both melano-cytes and RPE cells, similar to most of the other cell types in vertebrates, coopted the expression of evolutionary recent genes for effective incor-poration into the fast-evolving immune system framework.

## Discussion

In this study, we examine the plausibility and significance of the proposed homology between melanocytes, RPE, and ancient pigmented photosensory cells, particularly in the context of the evolutionary origin of the neural crest. Understanding the evolutionary emergence of the multipotent neural crest is fundamental to uncovering the origins of vertebrates and the diversifi-cation of body plans. The neural crest is the largest recent evolutionary innovation and a prime example of a multipotent cell type, capable of differentiating into a diverse array of cell types and tissues, now considered as a 4th germ layer, which is specific and defining for vertebrates[21,22]. Studies on the evolution of the neural crest often focus on identifying shared gene expression patterns between vertebrates and non-vertebrate chordates, aiming to reconstruct how the broad spectrum of neural crest-derived cell fates emerged gradually[72–80].

In our earlier work, we instead focused on a complementary evolu-tionary scenario, suggesting that the neural crest may have originated in protochordates in response to ecological pressures - specifically, the need for improved pigmentation and camouflage during a transition to active pre-datory lifestyle[3]. In brief, our evolutionary scenario underscores the homology of neural crest-derived melanocytes and other pigmented pho-tosensory structures found in vertebrates, tunicates and cephalochordates. The scenario itself suggests a developmental translocation of the neural

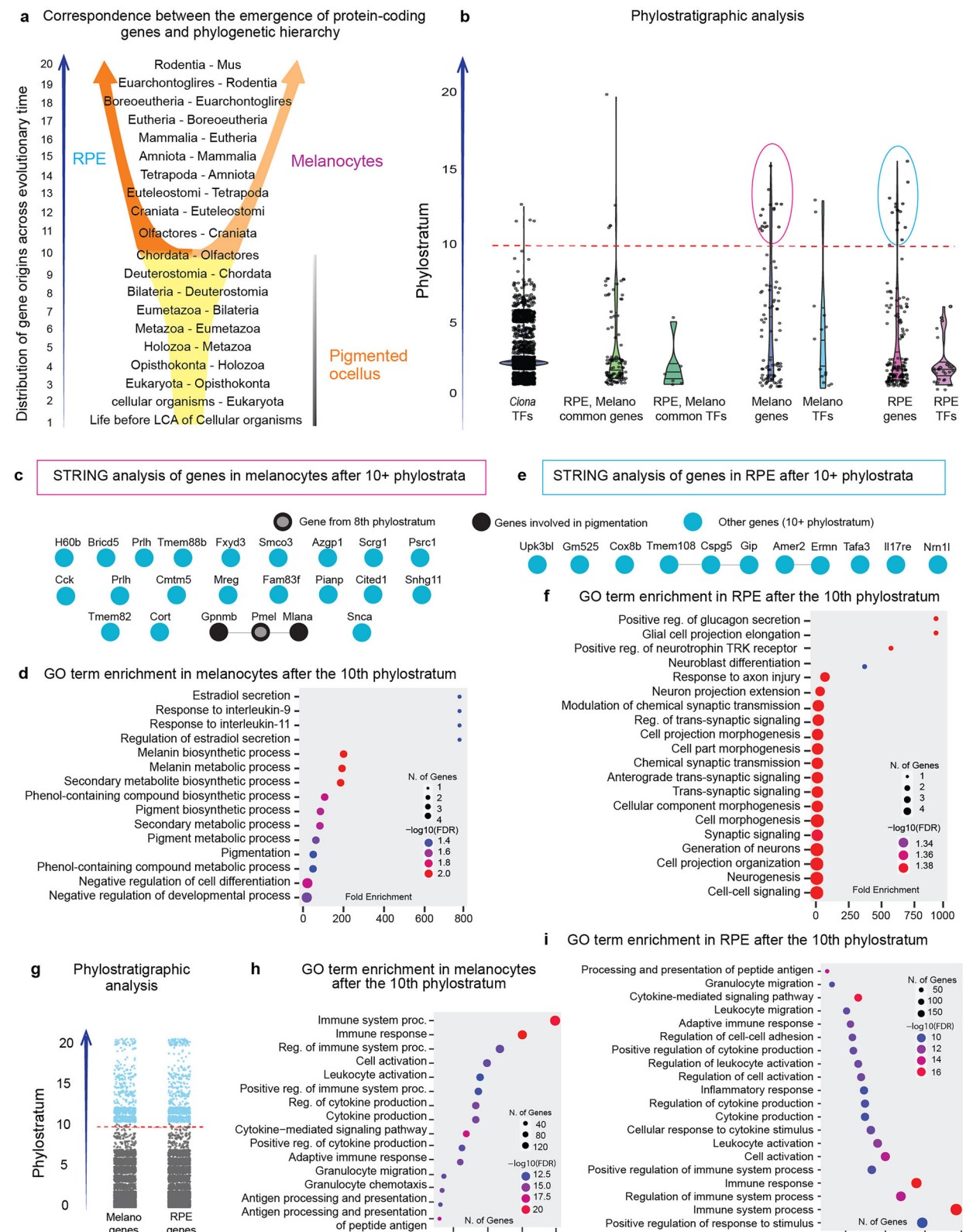

**Fig. 6 | Phylostratigraphy analysis of genes shared by melanocytes and RPE cells, as well as genes differentially expressed between them. a** Schematic representation of phylostrata corresponding to key evolutionary transitions. **b** Distribution of RPE- and melanocyte-enriched genes according to their evolutionary origin. **c, d** STRING (**c**) and GO term enrichment analysis (**d**) of melanocyte genes, which originated around 10th phylostratum and onwards.

**e, f** STRING (**e**) and GO term enrichment analysis (**f**) of RPE genes with emerged around 10th phylostratum and onwards. **g** Phylostratigraphy of all genes expressed in melanocytes and RPE in a non-cell type-specific manner, showing those originating from the 10th phylostratum and onward (highlighted in blue). **h, i** GO term enrichment of such genes in melanocytes (**h**) and RPE cells (**i**).

tube's pigmented photoreceptive cells to subcutaneous locations in ancestral protochordates, driven by the necessity to adapt to a larger body size and new ecological and environmental pressures[3] (Fig. 7). Initially, similar to the situation in amphioxus, the lateral intra-CNS pigmented ocelli (comparable to Hesse ocelli) could control the degree of hiding in sessile protochordates (as suggested by Lacalli) by sensing the incoming direction of light along the entire length of a body, as the pigmented ocelli were dispersed within the neural tube from head to tail[3,18,81].

As early chordates likely increased in body size prior to the evolution of active swimming and predatory behaviors[21,82], their deeply-located ocelli – simple pigmented light-sensitive structures within the neural tube—may have become increasingly shaded by expanding surrounding tissues. This reduction in light exposure could have exerted selective pressure for the relocation of photosensory function toward more exposed peripheral regions. We propose that this shift may have been facilitated by the emigration of specified neuroepithelial progenitors from the neural tube via epithelial-to-mesenchymal transition, enabling the establishment of pigmented ocelli in subcutaneous locations where light access was more reliable. As these animals evolved more complex predatory strategies, the anterior region began to specialize in frontal photosensory input[83–85], ultimately giving rise to true eyes. These newly centralized visual organs incorporated both optically shielding RPE and pigmented photoreceptors, gradually obliterating the visual function of the dispersed subcutaneous ocelli. However, the peripheral pigmented photoreceptors did not disappear; instead, they acquired a new function. Retaining their photosensitivity, they became dynamically light-responsive melanocytes, capable of regulating pigmentation in response to ambient light for purposes such as camouflage. These light-sensitive melanocytes thus represent a likely evolutionary offshoot of the broader lineage of pigmented photoreceptors. Their shared ancestry links them to the ancient pigmented photoreceptors and RPE cells of the vertebrate eye, suggesting an evolutionary relationship among pigment-producing, light-responsive cell types.

What, then, about the other neural crest-derived cell types within this evolutionary framework? It is plausible that the progenitors giving rise to melanocytes were already multipotent at the time of their emigration from the neural tube[60], allowing them to generate not only pigment cells but also other neural crest derivatives such as peripheral sensory neurons and glia. This idea is supported by the observation that the neural crest frequently generates cell types that are identical to those produced by other progenitor populations both within the CNS. A well-known example of such functional duplication involves Rohon-Beard neurons, somatosensory neurons that remain within the spinal cord, extend axons into the periphery, and exhibit diverse sensory modalities. These neurons co-exist alongside neural crest-derived dorsal root ganglion (DRG) neurons, which fulfill comparable sensory roles, and this overlap can persist for extended developmental periods, if not indefinitely[74,86]. This redundancy suggests that the emergence of neural crest derivatives was not strictly about inventing novel cell types, but rather about repositioning existing functions (including light-responding pigmentation) into new anatomical and ecological contexts - outside the CNS - through the mobilization of multipotent progenitors.

A central question in this context is what evolutionary and ecological pressures drove the emergence of the neural crest, and which of its diverse derivatives represent ancestral functions versus later co-opted fates. Our core hypothesis posits that vision and pigmentation played a foundational role in the origin of the neural crest. We argue that these functions - particularly through light-responsive pigment cells -offered early selective advantages that could have favored the evolution of migratory, multipotent progenitors. In contrast, alternative candidate cell types such as sensory neurons or glia appear less likely to have served as primary drivers. This is supported by the existence of Rohon-Beard neurons, which fulfill peripheral sensory functions while remaining within the CNS, as well as the mesodermal properties of ectomesenchymal cells, which suggest a later integration into the neural crest lineage. Thus, the emergence of neural crest identity may have initially been rooted in light-sensitive pigment cells, which implies that the modern melanocytes and pigmented cells of the visual

system (such as RPE and pinealocytes) represent the homologous cell types. With the recent development of single-cell atlases across both vertebrate and tunicate embryogenesis, as well as the emergence of powerful tools for comparative single-cell population analysis[42], we are now in a position to test this hypothesis using high-resolution, systems-level data.

To critically assess our central hypothesis - that melanocyte-RPE similarities reflect a homology - we considered an alternative explanation: that the observed transcriptional convergence might instead result from independent co-option of a pigmentation gene program by otherwise unrelated cell types. Under this scenario, melanocytes and RPE cells could share expression of pigment-related genes without being homologous. To evaluate this possibility, we conducted a deeper analysis of the shared gene expression profiles at single cell level. Importantly, we found that the similarities of RPE and melanocytes extended well beyond pigmentation genes, encompassing a broad range of functional modules related to metabolism, bioenergetics, and cellular maintenance. The presence of such diverse and functionally integrated gene networks shared between melanocytes and RPE cells, makes the hypothesis of broad co-option less plausible. Indeed, as established in evolutionary developmental biology, functional co-option typically involves the selective recruitment of a limited subset of a gene regulatory network into a new cellular context[87,88]. In contrast, the extensive overlap we observed suggests not a piecemeal appropriation of function, but a shared evolutionary origin.

We further decided to infer the evolutionary relations via investigating the similarity of pools of DNA-binding proteins in RPE and melanocytes: for this we performed the cladistics analysis based on the presence or absence of transcription factors and other DNA binding proteins, which highlighted melanocytes and RPE as the most related cell types in the anterior eye chamber. Cladistics approach encodes traits simply as present or absent and considers the distribution of such traits to determine the phylogenetic structure[40]. Importantly, cladistics analysis is specifically designed to reconstruct evolutionary histories, whereas hierarchical clustering groups data based on similarity with a number of drawbacks[38,89]. The cladistics approach to DNA-binding proteins (most of which are transcriptional controllers) simplifies the comparisons of cell types with complex gene expression profiles, where the difference in gene expression levels can strongly affect the results during computing the distance or dissimilarity between clusters. Here, the cladistics analysis based on transcription factor spectra recapitulated known phylogenetic relationships across diverse retinal and immune cell types, reinforcing the utility of this approach. Overall, both cladistics analysis based on expression of DNA-binding proteins and hierarchical clustering using global gene expression consistently positioned RPE cells and melanocytes as close neighbors among a wide range of cell types. Further comparative analysis of transcriptional modules regulated by specific transcription factors (regulons) revealed a significant overlap in transcriptional control between RPE and melanocytes, reinforcing the hypothesis of a shared evolutionary origin.

Given this substantial overlap in DNA-binding proteins, we extended our investigation to the underlying genomic regulatory landscape. Using a high-resolution 10x Multiome dataset from the human eye, which combines ATAC-seq and single-nucleus RNA-seq from the same cells, we examined chromatin accessibility profiles. This analysis revealed a broad set of shared accessible regions, particularly in non-promoter regulatory elements, between RPE and melanocytes. Notably, these two cell types clustered closely with Schwann cells in chromatin accessibility-based dendrograms - a result consistent with the known developmental derivation of melanocytes from Schwann cell precursors.

To extend this comparison beyond vertebrates, we analyzed single-cell transcriptomic data from the tunicate *Ciona intestinalis*[59], focusing on pigmented photosensory cells within the sensory vesicle. We then compared their transcription factor expression profiles to those of RPE and melanocytes in the mammalian anterior eye dataset. Strikingly, the pigmented photoreceptors of *Ciona* clustered with both RPE cells and melanocytes, sharing a substantial portion of their transcription factor code. The cells of the sensory vesicle from *Ciona*[90], as well as RPE and melanocytes shared the

**a**   Suggested scenario of the melanocyte origin via emigration of neural crest-like progenitors

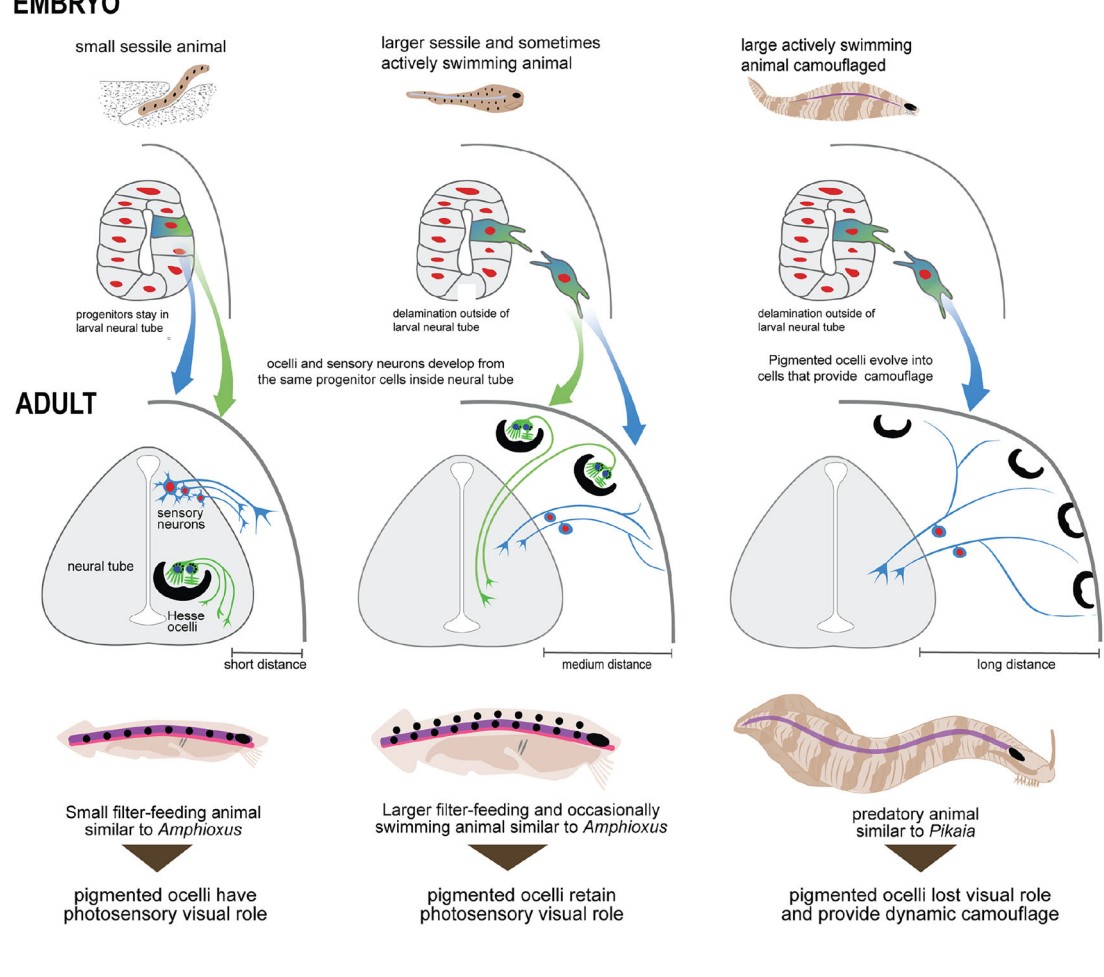

**b**   Evolutionary diversification of pigmented photosensory cell types in chordates

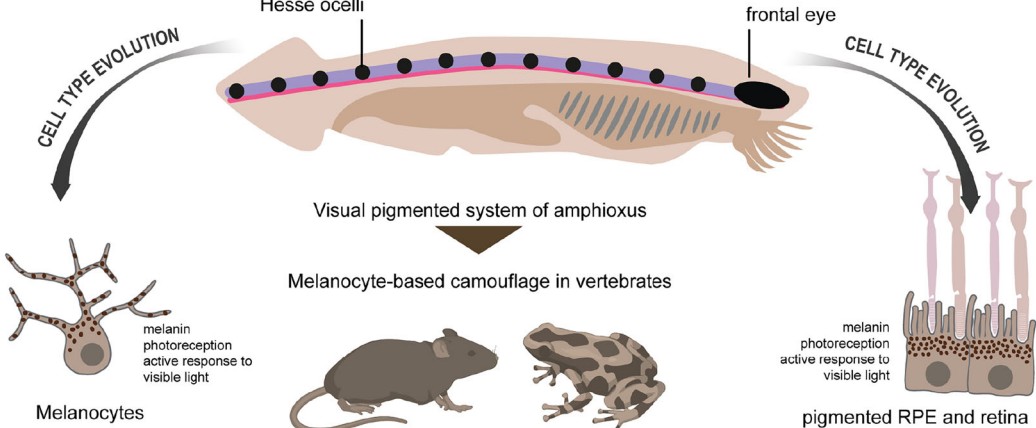

**Fig. 7 | Central hypothesis: evolution of the neural crest is linked to homology of RPE and melanocytes. a** Suggested scenario of the neural crest origin based on EMT of the CNS progenitors of pigmented ocelli for their placement in subcutaneous tissues and subsequent evolutionary transformation into photosensory pigmented melanocytes. **b** Scheme of evolutionary diversification of melanocytes and RPE cell types from the common origin of pigmented photoreceptors residing inside of the ancient chordate CNS. Created with BioRender.com. Licensed content used with permission.

expression of transcription factor *Mitf*, which is directly connected to a melanin pigmentation program by controlling the production of key enzymes involved in melanin synthesis, such as tyrosinase, tyrosinase-related protein 1, and dopachrome tautomerase[91]. MITF is associated with the pigmentation program in photosensory organs in quite distant animals such as amphioxus[5] and cubozoan jellyfish[2], which supports the ancient and archetypical role of MITF in melanin-based pigmentation in ocelli and eyes.

To test if *Mitf* and other pigmentation-related components will come up in other photosensory structures in vertebrates, also in association with melanin, we turned to the analysis of single cell data of the vertebrate "third eye", also known as a pineal gland[10]. The pineal gland has evolved significantly over time, transitioning from a light-sensing organ to a complex neuroendocrine structure producing melatonin and regulating the physiology of a day and night cycle. Pinealocytes are evolutionary derived from photoreceptors and they still express a number of opsins and elements of a phototransduction cascade[9,10]. As a result of exploring the single cell data, we found the robust presence of *Mitf* and much weaker presence of *Sox10* and *Tyr* in pinealocytes. The low expression levels of *Sox10* and other pigmentation related genes might explain the weak melanization of pineal gland[8], as compared to completely dark RPE and melanocytes.

Encouraged by the observed similarities, we next examined the nature and developmental timing of differences between melanocytes and RPE cells. We detected late expression of RPE-specific genes (vs melanocyte-specific genes) during the development of RPE, whereas the melanocyte-like pigmentation program appeared in developing RPE much earlier, thus rendering early developing RPE cells as more similar to melanocytes. Furthermore, RPE-specific programs were enriched for genes involved in epithelial organization, molecular transport, and the retinol/retinal cycle, while melanocyte-specific programs reflected mesenchymal morphology and interactions with keratinocytes. These differences are expected, reflecting lineage-specific adaptations: RPE cells support photoreceptor function through metabolic and biochemical pathways[35], whereas melanocytes transfer pigment granules to keratinocytes in skin, hair, and feathers for camouflage and UV protection[92]. Additionally, melanocytes have evolved structural and metabolic roles in tissues such as the inner ear and heart[93–95].

The nature and functional significance of late evolutionary acquisitions can be further explored using phylostratigraphy, a method that assigns genes to evolutionary strata based on their origin and partitions the transcriptome accordingly[65,66]. Applying phylostratigraphy to differentially expressed genes between melanocytes and RPE, we found that melanocytes gradually acquired genes central to melanosome transfer to keratinocytes (*Pmel*, *Gpnmb*, and *Mlana*). In RPE cells, recently evolved genes included those related to barrier function (*Cdh3*), transport - likely of melanin precursors (*Oca2*) - and endocrine signaling (*Tshr, Ins2, Gip, Npr1*), aligning with the RPE's integrative role in photoreceptor support and retinal insulation. We also examined recently evolved genes expressed in both RPE and melanocytes but not specific to either. Most were linked to immune interactions, particularly histocompatibility and surveillance. This aligns with evidence that the adaptive immune system underwent rapid evolution in vertebrates, requiring all cell types, including pigment cells, to engage in immune monitoring and defense against infection and malignancy[70,71].

Overall, despite the functional diversification of RPE and melanocytes, our analysis reveals extensive shared features beyond melanin production, including photoreception, metabolism, energetics, and biochemical support, as well as overlapping transcriptional control. These findings support the hypothesis of ancient homology among melanocytes, RPE, and possibly pinealocytes. Comparative transcription factor analysis between vertebrate pigment cells and tunicate pigmented ocelli further strengthens this idea. This implies that the pigmentation program was not coopted into emerging melanocytes, and instead, melanocytes were one of the very first fates driving the origin of the multipotent neural crest.

At last, we would like to mention the limitations of this study. In evolutionary biology research, evidence is most often suggestive rather than direct and conclusive. As a result, researchers rely on indirect evidence, such as similarities in gene expression profiles, regulatory elements, or cell types

between distantly related species. This study is no exception. Furthermore, comparative evolutionary single-cell biology is still in its developmental stages, and key datasets remain unavailable - for example, single-cell transcriptomics covering amphioxus photosensory organs and cyclostome pigment systems. Therefore, this study should be regarded as a foundation for future work rather than definitive proof of homology between melanocytes and RPE. Further progress will require a robust evolutionary framework, including genome-wide regulatory landscape analyses, which are not yet fully achievable.

## Methods

### Technical procedures for Fig. 1, Notebook 1a, 1b
In this study, we reanalyzed single-cell RNA sequencing data from the Gene Expression Omnibus (GEO) repository, specifically sample GSM5560840, including RPE. We opted for the Seurat package (v5.0.3) in R (version 4.2.0)[96] for data analysis, as detailed in Fig. 1, *Notebook 1_a*. We selected only cells with 500–20,000 UMIs (Unique Molecular Identifiers), 800–6000 genes, log10(genes/UMI) > 0.8, and mitochondrial gene percentage < 10%. Seurat normalization was followed by selection of 3,000 variable genes. These were scaled, and PCA (Principal Component Analysis) was performed on 30 PCs (Principal Components) for clustering. Next, we took 30 PCs to build the nearest-neighbor graph, perform Leiden clustering[97] and visualize results with UMAP (Uniform Manifold Approximation and Projection)[98]. Potential doublets were identified and removed using the DoubletFinder algorithm[99]. The estimated doublet rate was ~7% among all cells. We applied DoubletFinder focusing on the top 10 principal components. Identified doublets were excluded from the downstream analyses. After the doublet removal procedure, the clustering and dimensionality reduction processes were repeated. PCA was recalculated, focusing on the first 20 principal components. Clustering resolution was set to 0.1, and then UMAP was re-run.

Next, we identified cell types based on known cluster-specific marker genes. RPE and Melanocyte cells were isolated computationally (Fig. 1, Notebook 1_b), then sub-clustered using 20 PCs at resolution 0.01, and visualized by UMAP. Next, we identified key marker genes in the RPE and Melanocyte sub-clusters using Seurat's FindAllMarkers function[96]. Genes were selected based on a minimum expression in 25% of cells and a log fold change threshold of 0.25. Marker genes identified in the RPE and Melanocyte clusters were further analyzed through protein-protein interaction (PPI) analysis using the STRING database (version 11.5)[100]. The STRING (Search Tool for the Retrieval of Interacting Genes/Proteins) tool is an online resource used to analyze protein-protein interactions and functional associations from gene lists. It integrates information from various sources, including experimental data, computational predictions, and public text collections, to build interaction networks. This analysis revealed potential interactions between proteins encoded by identified marker genes, offering insights into the molecular networks and pathways active in these cell populations. Additionally, we conducted Gene Ontology (GO) enrichment analysis using the ShinyGO tool (version 0.75)[101], providing further insights into the biological processes and pathways that define these cell types.

### Technical procedures for Fig. 2, Notebook 2
Figure 2, *Notebook 2* describes analysis of genes enriched in the RPE and Melanocyte clusters as compared to other cell populations. We took the entire embedding from Fig. 1, *Notebook1_a*, then merged RPE and Melanocyte clusters, and identified highly variable genes using the FindAllMarkers function[96]. Genes with avg. logFC >3 were selected for further analysis. Next, we visualized GO-term enrichment scores using ShinyGO tool (version 0.75)[101] and performed protein-protein interaction analyses with STRING (version 11.5)[100] to identify dominating biological processes and key molecular pathways.

### Technical procedures for Fig. 3, Notebook 3_a, 3b
For the DNA-binding proteins (transcription factor) analysis, we took advantage of the dataset from Fig. 1. The parameters for quality control were

the same as in *Notebook1_a*. Next, we opted for the standard list of mouse DNA-binding proteins (mostly transcription factors) from the Animal Transcription Factor Database (AnimalTFDB)[102], (Fig. 3, *Notebook 3_a*) for building the new embedding. PCA was performed using 40 components, followed by the re-clustering and UMAP visualization. A dot plot visualized the top six marker genes for each cell type. A heatmap was generated to highlight the top transcription factor markers for each cluster. Seven genes per cluster were selected based on their highest average log2 fold change, Supplementary Fig. 1. Hierarchical clustering was performed using the hclust function, and the resulting dendrogram was visualized to explore relationships between the clusters[103].

For parsimony-based cladistics tree analysis corresponding to *Notebook 3_b*, we prepared a character matrix storing the transcriptional factor presence data for each cell[103]. For each cluster, we compared genes to a reference list of mouse DNA-binding proteins (most of which are TFs). The specific DNA-binding protein/TF presence was recorded as binary (1 = present, 0 = absent). To reconstruct lineage relationships between cells, we used the CASSIOPEIA toolkit (Jones et al., 2020), *Notebook 3_c*. CASSIOPEIA is a scalable platform that typically reconstructs phylogenetic trees from CRISPR-Cas9 lineage tracing data, but it can also be adapted for transcriptional factor presence/absence datasets, as in our study. By applying parsimony-based methods, CASSIOPEIA allowed us to infer the potential evolutionary relationships between cell clusters using cladistics/parsimony approach. To build the parsimony-based tree, we used CASSIOPEIA's Integer Linear Programming (ILP) solver, which seeks to minimize the number of changes (mutations or shifts in gene expression) required to explain the observed data. The ILP solver iteratively assessed possible tree structures, optimizing for the most parsimonious solution. Critical parameters, such as the convergence time limit and graph layer size, ensured efficient tree reconstruction without compromising accuracy. Once the tree was generated, we visualized the relationships between cell clusters using Interactive Tree Of Life (iTOL)[104], an online tool for tree visualization.

### Technical procedures for Notebook 3_d, 3_e

To reconstruct the gene regulatory networks (GRNs) in RPE cells and melanocytes, the SCENIC pipeline was applied to scRNA-seq data from Fig. 1[48,105], Fig. 3, *Notebook 3_d*. First, we identified genes linked to transcription factors using GRNBoost2. Then, we confirmed these links using DNA motif analysis with cisTarget. Finally, we measured how active each gene set (regulon) was in individual cells using AUCell. In total, we found 387 regulons in 16,227 cells. The results were added to an AnnData object and visualized with dot plots and UMAPs using Scanpy (Notebook 3_e).

SCENIC identifies regulatory programs by integrating gene co-expression and transcription factor (TF) motif enrichment analysis. The analysis involved the following steps:

1. GRN Inference: Using the GRNBoost2 algorithm, co-expression patterns between transcription factors and target genes were inferred. This step generated an adjacency matrix representing transcriptional regulatory interactions.
2. Regulon Definition: The TF-target relationships were refined through motif enrichment analysis using the cisTarget databases. This analysis identified direct TF targets based on conserved binding motifs around transcription start sites, leading to the definition of regulons.
3. Transcription Factor Activity: The AUCell algorithm was used to score regulon activity within individual cells, quantifying transcription factor activity across the RPE and other cell populations.

The SCENIC results were integrated into an AnnData object, including 16,227 cells and 387 regulons. Visualizations in Scanpy (version 1.9.3)[106] such as dot plots and UMAP were generated to show regulon activity across different cell types, providing insights into transcriptional programs specific to RPE and melanocyte populations, *Notebook 3_e*.

### Technical procedures for Fig. 4a, b, Notebook 4_a

Here we reanalyzed scRNA-seq data from GEO (GSE155121), using Scanpy (version 1.9.3)[106], focusing on developmental stages from weeks 4 to 8 (Fig. 4, *Notebook 4_a*). Quality control was performed separately for each stage. For Week 4, cells were included if they had between 1000 and 6000 genes, mitochondrial content below 6%, and fewer than 30,000 total counts. In Week 5 dataset, cells with 1000 to 4000 genes, mitochondrial content below 6%, and total counts under 15,000 were retained. Week 6 cells were selected if they expressed 1000 to 6000 genes, had mitochondrial content below 5%, and total counts between 5000 and 30,000. For Week 7, the criteria were 1000 to 5000 genes, mitochondrial content under 5%, and fewer than 20,000 counts. Lastly, Week 8 cells were filtered to include those with 1000 to 4000 genes, mitochondrial content below 5%, and fewer than 20,000 total counts.

Subsequent data processing included normalization and logarithmic transformation. In addition, we removed the cell cycle genes, which we took from Gene Ontology database[107], *Notebook 4_b*. Next, we identified highly variable genes, scaled the data, and performed PCA with 30 components. A neighborhood graph with 30 nearest neighbors was built, followed by UMAP for dimensionality reduction. Clustering was done using the Leiden algorithm[97] at resolution 0.5.

### Technical procedures for Fig. 4c, d Notebook 4_c

In this part, we reanalyzed scRNA-seq data from GEO (GSE210543) using Scanpy (version 1.9.3)[106]. Cells with 1500–6500 genes and mitochondrial gene expression below 5% passed QC. After normalization and log transformation, we opted for 3000 variable genes. The data were scaled, and PCA was performed using the arpack solver. Next, the top 20 PCs were used to build a neighbor graph, apply UMAP, and run Leiden clustering at resolution 0.5. Clusters expressing *RPE65, PMEL, TYR,* and *DCT* were kept for sub-clustering. After restoring the raw data and selecting new variable genes, we scaled the data and re-run the PCA. A new neighbor graph (30 neighbors, 10 PCs) was used for UMAP and Leiden clustering (res = 0.5), identifying RPE and Melanocyte clusters.

### Technical procedures for Fig. 4, Notebook 4_d(1), 4_d(2)

We reanalyzed Stereo-seq data from the CNGB Nucleotide Sequence Archive (accession code: CNP0001543), focusing on two developmental stages: E14.5 (containing 105,857 cells and 29,245 genes), *Notebook 4_d(1)*, and E16.5 (containing 73,922 cells and 27,762 genes), *Notebook 4_d(2)*. Since these datasets had already passed quality control, we processed each developmental stage separately using Scanpy[106]. We normalized the data to adjust for differences in total cell counts and log-transformed it to reduce expression variance. We performed PCA to capture major variation, built a nearest-neighbor graph, and applied UMAP for visualization. For spatial analysis, we used the Squidpy package[108] to explore patterns. Then, we performed clustering using the Leiden algorithm[97] optimized to detect well-connected cellular communities. Additionally, cell-type-specific scores were calculated for melanocytes and RPE cells using known marker genes, confirmed by spatial distribution and identity of these cell types.

### Technical procedures for Fig. 5

We downloaded single-cell RNA sequencing data from GEO (GSE131155)[59]. This dataset includes 15 samples collected from various developmental stages of the *Ciona intestinalis* specifically: Initial Tailbud I (iniTI): 2 samples, Early Tailbud I (earTI): 2 samples, Mid Tailbud II (midTI): 2 samples, Late Tailbud I (latTI): 6 samples, and the Larval stage: 3 samples (Fig. 5a, *Notebook5_a*). Raw gene-barcode matrices for each sample were downloaded in HDF5 format and processed using the Scanpy package (v1.8.2) in Python (v3.8.15)[106]. The samples were later merged. As a result, we obtained a dataset with 11,059,200 cells and 15,269 genes. Cells expressing <300 genes or >6500 genes were excluded during quality control, as they were likely of low quality or doublets. Than we normalized dataset by scaling the total gene expression of each cell to 10,000 counts, followed by log-transformation to stabilize variance across genes[109].

We performed feature selection by identifying 1000 most highly variable genes, which were used in the subsequent analyses. To ensure each gene contributed uniformly to downstream analyses, gene expression was scaled. We next built PCA using default parameters[97]. A neighborhood graph was constructed using 35 principal components and 45 nearest neighbors to capture local data structure, and UMAP was applied to visualize the data in two dimensions[98]. Next, we performed clustering using the Leiden algorithm at resolution of 0.7, identifying distinct cell populations and visualizing them with UMAP.

Cluster 38 appeared to be enriched for RPE-related genes, including *Dct*, *Tyr*, and *Tyrp1*. We applied Scanpy's rank_genes_groups function with the *t*-test method to identify marker genes. The top-ranked genes were organized into a DataFrame for further analysis. Genes with *p*-values < 0.05 and log fold changes >3.5 were selected for detailed investigation. To facilitate cross-species comparisons, *Ciona intestinalis* genes were converted into their mouse orthologs using gene conversion tables obtained from Singh-Lab GitHub (2020), *Notebook5_b*. The differentially expressed genes were analyzed for protein-protein interactions (PPIs) using the STRING database[100]. Additionally, we opted for Gene Ontology (GO) enrichment analysis with ShinyGO tool to explore the biological processes and pathways associated with the differentially expressed genes[101] (Supplementary Fig. 7)

To construct a parsimony-based tree with the CASSIOPEIA toolkit[42], we used the character matrix from *Notebook 3_d* and added a new column containing *Ciona* genes generated in *Notebook 5_b*. The CASSIOPEIA parameters were kept identical to those used earlier, and the tree was computed and visualized using the same settings, *Notebook5_c*.

For Fig. 5e, the adult *Ciona intestinalis* type A (Pacific species, also called *Ciona Robusta*) were collected by Marinus Scientific in Long Beach, California. The animals are hermaphrodite invertebrates and do not require ethical committee oversight. The animals were maintained in artificial seawater at 18 °C under constant illumination until gamete collection. Sperm and eggs were isolated from the sperm duct and oviduct, respectively. For fertilization, sperm from one animal was mixed with chorionated eggs of another animal for 10 min. The eggs were washed three times in artificial seawater before being reared at 18 °C in artificial seawater. Stage 27 to 28 larvae[110] were fixed using the fixative published by Treen and colleagues[111] and composed of 100 mM HEPES, 500 mM NaCl, 1.75 mM MgSO4, 2 mM ethylene glycol bis (succinimidyl succinate) (Thermo Fisher Scientific, 21565), and 1% formaldehyde (Thermo Fisher Scientific, 28908) under constant agitation overnight at 4 °C. They were then washed four times in PBST and dehydrated in increasing concentrations of Ethanol to reach 80% Ethanol. They were stored at -20 °C in 80% Ethanol until further use. Larvae were collected over three different crosses. The mRNA of *Hh2*, *Otx*, *Rpe65*, *Rrh*, and *Tyr* (identifier KH.C8.537 and KH.C5.485, KY21 and KH2012 identifiers in Table) was detected by HCR with probes from Molecular Instruments using their HCR™ Gold RNA-FISH kit. Larvae were rehydrated in 50% Ethanol, followed by 25% Ethanol, 50% phosphate buffer saline with 0.1% tween (PBST), and four washes in PBST. They were then blocked in 100 μl probe hybridization buffer for 2 h at 37 °C before 100 μl of probes diluted in the probe hybridization buffer was added to a final concentration corresponding to the recommended manufacturer dilution. Larvae were hybridized overnight at 37 °C. The next day, 200 μl probe wash buffer was added to allow larvae to sink at 37 °C. They were washed four times at 37 °C in 200 μl probe wash buffer as described in the manufacturer protocol. The larvae were then pre-incubated in 50 μl of amplifier buffer for 1 h at room temperature. The amplifier buffer was discarded, and the larvae were then incubated overnight at room temperature in 50 μl of amplifier solution with the desired hairpin amplifiers prepared as described by the manufacturer. The next day, this solution was collected. It can be stored at -20 °C and reused at least three times. Before utilization, it was heated for 5 min at 95 °C before cooling at room temperature for at least 30 min. The larvae were washed four times in amplifier wash buffer as described by the manufacturer, followed by three washes in PBST, stained with the nuclear dye 1 μg/ml Dapi in PBST for 30 min, and washed three times in PBST. The

larvae were finally mounted on slides with FluorSave (Sigma, 345789) before being imaged with a Leica Sp8 scanning confocal microscope (image acquisition software LAS X version v3.5.7.23225). The images were analyzed with Fiji ImageJ2 (version 2.14.0/1.54j)[112].

**Table of gene nomenclature**

| Gene name | KH2012 Identifier | KY21 identifier | Closest human gene | Synonyms |
|---|---|---|---|---|
| Hh2 | KH2012:KH.C5.544 | KY21.Chr12.575 | IHH | Hh.b, Hedgehog2 |
| Otx | KH2012:KH.C4.84 | KY21.Chr4.720 | OTX1 | Otx1/Otx2 |
| Rpe65 | KH2012:KH.C9.224 | KY21.Chr9.932 | BCO1 | |
| Rrh | KH2012:KH.L57.28 | KY21.Chr12.157 | RGR | Opsin3, Rgr |
| Tyr | KH2012:KH.C8.537 | KY21.Chr8.260 | DCT | Dct/Tyr, Tyrp1/2a, Tyrp1b |
| Tyr | KH2012:KH.C5.485 | KY21.Chr5.389 | TYRP1 | Dct/Tyr, Tyrp1a |

Names of the genes investigated by HCR with their identifier from the KH2012 gene model and KY21 gene model, their closest human gene, as well as commonly used synonyms.

### Technical procedures for Fig. 6

For the phylostratigraphy analysis, we used the phylogenetic map developed by Šestak and coauthors[113], which encompasses 20 phylostrata. Each phylostratum represents a significant evolutionary transition, ranging from the Last Common Ancestor (LCA) of all cellular organisms to *Mus musculus*. This phylogenetic map was applied to assign the time of the evolutionary origins to all individual genes in our dataset. Gene annotations for *Mus musculus* were taken from the Ensembl genome browser (release 109). The dataset included Ensembl gene IDs and mouse gene names, which were downloaded and processed using R (version 4.2.0) (Fig. 6b, *Notebook 6*). The genes were mapped to their respective phylostrata based on the evolutionary map described above. Gene expression data for the analysis of clusters were obtained from the main embedding shown in Fig. 1[55] and analyzed in Seurat (version 5.0.3), as detailed in *Notebook 1*. Next, we performed differential gene expression analysis using Seurat to identify marker genes for melanocytes and RPE. For melanocyte and RPE markers, we applied the following threshold *p*-value < 0.05 and log2 fold change (log2FC) > 3.5 to ensure the statistical significance. To visualize the expression profiles of RPE and melanocyte-specific genes, we generated a heatmap using Seurat (version 5.0.3), Supplementary Fig. 1c, d. The identified marker genes for both melanocytes and RPE were integrated with the Ensembl dataset to assign phylostratum levels. Genes starting at the phylostratum levels 10 and above were selected for further analysis.

### Technical procedures for Supplementary Fig. 2a–d, Notebook 7a, 7b

In the following analysis, we selected RPE and melanocyte clusters from the anterior eye chamber single cell dataset from (GSM5560840, GEO / (Lee et al., 2022)[33]) and merged them with the mouse back skin data (GSE131498, GEO / (W. Ge et al., 2020)[43]) from postnatal day 0, (*Notebook 7a*). All datasets were processed using Scanpy (v1.9.1)[106] in Python (v3.8.10). Quality control measures included retaining genes expressed in at least three cells and cells with a minimum of 200 detected genes. To correct for the differences in sequencing depth, total counts per cell were normalized to 10,000, followed by log transformation. We assigned a batch label to each dataset to facilitate batch correction in subsequent steps. We selected the top 2000 highly variable genes for embedding and downstream analysis[96], and the data were scaled. Next, we run PCA using 120 components. To correct batch

effects, we applied Harmony (v0.1.0)[114] to adjust PCA embeddings. Next, a KNN graph was built using the top 35 Harmony-corrected PCs. Then, we used Leiden clustering to define cell groups, and UMAP was applied for visualization. Clusters were annotated based on marker gene expression, identifying RPE, melanocytes, and other populations. Differential expression analysis was performed to compare transcriptional programs between cell types. To investigate the evolutionary relationships between RPE, melanocytes, and additional cell types, we applied hierarchical clustering based on pairwise correlation distances calculated from the mean gene expression profiles of each cell type. A dendrogram was generated using Ward's linkage method.

In parallel, we constructed a parsimony-based tree using the CAS-SIOPEIA tool[42], TFs expressed in at least 10% of cells within each cluster were identified, and a binary presence/absence matrix was generated. We selected TFs, which are expressed in at least one but not in all clusters, allowing for the identification of lineage-specific regulatory factors. This matrix was further used for the evolutionary inference in CASSIOPEIA-based analysis[42,115], following standardized table preparation methods. Once the phylogenetic tree was generated, we used Interactive Tree of Life (iTOL)[104] for visualization, *Notebook 7b*.

## Technical procedures for Supplementary Fig. 2e–h, Notebook 8a, 8b

In this study, we selected RPE and melanocyte clusters from the anterior eye chamber single cell dataset from (GSM5560840, GEO/(Lee et al., 2022)[33]) and merged them with the mouse back skin data (GSE181390, GEO / (Lee et al., 2024)[44]) from postnatal day 4, *Notebook 8a*. All datasets were processed using Scanpy (v1.9.1)[106], Python (v3.8.10). Quality control measures included keeping genes that were expressed in at least three cells and selecting cells with at least 200 detected genes. To correct for differences in sequencing depth, total counts per cell were normalized to 10,000, followed by log transformation. A batch label was assigned to each dataset to facilitate batch correction in subsequent steps. Datasets were integrated by concatenating them while aligning common genes. Non-informative metadata columns were removed, and the 2000 most highly variable genes were selected for downstream analyses. The dataset was then scaled, and PCA was performed, retaining the top 50 components. To address batch effects, Harmony (v0.1.0)[114] was applied, adjusting PCA embedding to ensure alignment of cellular identities across datasets.

For the cell type annotation and clustering, a K-nearest neighbor (KNN) graph was constructed using top 35 Harmony-adjusted PCA components. Leiden clustering was applied to define cell populations, while UMAP was used to visualize the cellular landscape. Cluster identities were determined based on marker gene expression, enabling the classification of RPE, melanocytes, and other distinct cell populations. Differential gene expression analysis was performed to compare transcriptional programs among cell types. Hierarchical clustering and a parsimony-based tree were constructed as previously described for Supplementary Fig. 2b, c, *Notebook 8b*.

## Technical procedures for Supplementary Fig. 2i–l, Notebook 9a, 9b

Here, we selected RPE and melanocyte clusters from the anterior eye chamber single cell dataset from (GSE131498, GEO/(W. Ge et al., 2020)[43]) and merged them with activated melanocyte stem cells from anagen II hair follicles of mouse back skin (GSE203051, GEO / (Sun et al., 2023)[45]) at 5 weeks of age, *Notebook 9a*. This analysis followed the same workflow as described previously for Supplementary Fig. 2b, c, *Notebook 9b*. PCA was performed using the top 50 components. The K-nearest neighbor (KNN) graph was constructed using the top 35 Harmony-adjusted principal components. Leiden clustering was applied with a resolution of 0.5. Hierarchical clustering and the parsimony-based tree were constructed as previously described for Supplementary Fig. 2b, c, *Notebook 9b*.

## Technical procedures for Supplementary Fig. 2m–p, Notebook 10a, 10b

In this study, we selected RPE and melanocyte clusters from the anterior eye chamber single cell dataset from (GSE131498, GEO / (W. Ge et al., 2020)[43]) and merged them with Smart-seq2 data (GSE147298 / (Infarinato et al., 2020)[46]) from wild-type quiescent melanocyte stem cells in mice, *Notebook 10a*. All datasets were processed using Scanpy (v1.9.1) in Python (v3.8.10). Quality control measures included retaining genes expressed in at least three cells and cells with a minimum of 200 detected genes. To correct for differences in sequencing depth, total counts per cell were normalized to 10,000, followed by log transformation. A batch label was assigned to each dataset to facilitate batch correction in subsequent steps. Datasets were integrated by concatenating them while aligning common genes. Non-informative metadata columns were removed, and the 2000 most highly variable genes were selected for downstream analyses[96]. The dataset was then scaled, and Principal Component Analysis (PCA) was performed, retaining the top 100 components. To address batch effects, Harmony (v0.1.0)[114] was applied, adjusting PCA embeddings to ensure alignment of cellular identities across datasets. For cell type annotation and clustering, a K-nearest neighbor (KNN) graph was constructed using top 30 Harmony-adjusted PCA components. We then applied Leiden clustering[97] to define cell populations, while UMAP[98] was used to visualize the cellular landscape. Cluster identities were determined based on marker gene expression, enabling the classification of RPE, melanocytes, and other distinct cell populations. Next, we performed differential gene expression analysis to compare transcriptional programs among cell types. Overall, hierarchical clustering and parsimony-based tree were constructed as previously described for Supplementary Fig. 2b, c, *Notebook 10b*.

## Technical procedures for Supplementary Fig. 3, Notebook 11

Single-nucleus RNA sequencing (snRNA-seq) datasets for the pineal gland and RPE from macaque monkeys were obtained from the NHPCA database (https://db.cngb.org/nhpca/), generated using the DIPSEQ-T1 platform[116]. Samples were collected from eight 6 year-old *Macaca fascicularis*. Sequence alignment was performed using a custom *Macaca fascicularis* 5.0 "pre-mRNA" reference genome. Two Seurat objects were downloaded: one for the pineal gland (19,864 features across 14,501 nuclei) and another for the retinal pigmented epithelium (19,638 features across 12,091 nuclei)[53], *Notebook 11*.

The Seurat objects were processed and analyzed using the Seurat package (v5.0.3) in R (version 4.2.0)[96], as detailed in Figure supplementary 3, *Notebook_suppl3*, following standardized workflows[109]. Initially, the Seurat objects were imported into R, and cell annotations were transferred to new Seurat objects corresponding to the pineal gland and RPE datasets. These objects were merged for the integrated analysis. The top 2000 most variable genes were selected using the variance-stabilizing transformation (VST) method, followed by data scaling with ScaleData() in Seurat. Principal component analysis (PCA) was conducted using the RunPCA() function in Seurat.

For the cell clustering, we constructed a shared nearest neighbor (SNN) graph using the FindNeighbors() function, identifying the 20 nearest neighbors in PCA space, and the clustering was performed using FindClusters() with a resolution of 0.5. Visualization of the clusters was achieved using UMAP via the RunUMAP() function. After excluding non-informative clusters, re-clustering was performed using 15 nearest neighbors, followed by UMAP with the top 15 principal components. We assigned cell types based on the reference literature, resulting in the identification of astrocytes, endothelial cells, macrophages, melanocytes, microglia, pinealocytes, RPE, and stromal cells. The primary focus of the analysis was on pinealocytes, melanocytes and RPE, with other cell types categorized as "rest."

## Technical procedures for Supplementary Fig. 4, Notebook 12

For the comparison of chromatin states between RPE and melanocytes we used snRNAseq data for all control samples from the study under GEO

GSE202886[55], *Notebook 12*. First, we merged normalized RNA and ATAC matrixes into the combined dataset object using Seurat (v 5.0.0)[117] and Signac (v 1.14.0)[118] packages on R (v 4.4.2, R Core Team 2021), and performed following analysis using these packages. All unlabeled cell types were filtered out based on the metadata. We kept only peaks that were present at least in 1% of cells. FindTopFeatures() was used to identify most observed variable features (atac-seq peaks), following singular value decomposition RunSVD() for dimensionality reduction. Then, we performed RunU-MAP(dims = 2:30) for visualizing cell clusters (Supplementary Fig. 4a). Original annotations (RNA-seq-based) from the corresponding literature were used to annotate the identified cell types. Next, we computed the nearest neighbor graph FindNeighbors (dims = 2:30) and identified clusters of cells using FindClusters (resolution = 0.3) based on their ATAC-seq dimensionality reduction. 11 identified clusters of cells were renamed based on the majority of annotated cell types, which are contributing to those clusters of cells ('ATAC-seq' based cluster). We calculated proportion of cells with original RNA-based annotation in the newly defined ATAC-seq-based cluster using dplyr package (v 1.1.4)[119] and visualized results using ggplot2 package (v 3.5.1)[120] (Supplementary Fig. 4b). To find peaks that are correlated with expression of nearby genes (distance = 1,000,000) we used LinkPeaks() function. As an example, we visualized *Sox10* gene that had multiple regions correlating with its expression using CoveragePlot() (Supplementary Fig. 4d).

Prior to calculating the similarity trees on ATAC-seq assay, we additionally subset the dataset by excluding ATAC-seq peaks overlapping with all promoter regions (defined by promoters() function as 2000nt upstream and 200 downstream of TSS of the gene), and gene coordinates including introns. Then, we extracted LSI coordinates (2:30 components) and computed centroids on the mean of those LSI coordinates for each cell type. Next, we computed pairwise distance dist() between those centroids (for distance measure we opted for Euclidian, maximum, manhattan, and Canberra distances). Then we clustered the output using hclust() and built it as a dendrogram using as.dendrogram() functions (Supplementary Fig. 4c, 4e).

### Technical procedures for Supplementary Fig. 5, Notebook 12

To identify differentially accessible (DA) peaks that are common for melanocytes and RPEs (original RNA-based annotation) we used FindMarkers() function with build-in Wilcoxon test. We visualized the output using volcano plot with ggplot() (Supplementary Fig. 5a). Peaks were considered as DA if adjusted *p*-value threshold <0.05 and logFC >2 for upregulation and logFC <-2 for downregulation. The resulting table was filtered out for average logFC > 2 and adjusted *p*-value < 0.05. For those peaks we identified associated genes from LinkPeaks() output. We uploaded the resulting gene set to STRING[100] to visualize interactions and enriched Biological Processes (Gene Ontology) (Supplementary Fig. 5b, c).

### Technical procedures for Supplementary Fig. 6, Notebook 13

Next, we integrated the dataset from Li et al. 2024[56] with the anterior eye chamber dataset from Lee et al. 2022[33] using the Scanpy toolkit (v1.9.1), *Notebook 13*. To reduce dimensionality and focus on the informative features, we identified the top 2000 highly variable genes using the *Seurat* selection method. The data were scaled to unit variance with a maximum value of 10, and PCA was performed, retaining 60 components and using the ARPACK algorithm for decomposition.

To correct for batch effects between datasets, we applied Harmony integration in PCA space, specifying the sample origin as the batch variable. The integration parameters were set as follows: theta = 3.0 to adjust the integration strength, lambda = 0.5 to control regularization, max_iter_harmony = 3 to limit the number of iterations, and epsilon_cluster = 1e-3 to loosen clustering convergence criteria. The corrected embeddings were stored for downstream analysis. For visualization, UMAP was applied to the Harmony-corrected data. We used custom colormaps constructed from grey and red gradients for improving the contrast. The resulting UMAP plots were annotated with

known markers for RPE and melanocytes, including *Rpe65*, *Dct*, *Tyr*, *Mitf*, *Pmel*, and *Mlana*, along with Leiden cluster assignments and pre-annotated cell type identities to assess consistency across clusters. For the CASSIOPEA dataset, we used a binary matrix, following the same pre-processing steps as described above. For visualization, we used Interactive Tree of Life (iTOL)[104], an online platform for displaying and annotating phylogenetic trees.

### Reporting summary

Further information on research design is available in the Nature Portfolio Reporting Summary linked to this article.

### Statistics and reproducibility

All analyses were performed on publicly available datasets, reprocessed and reanalyzed using standardized, well-documented computational pipelines. Cell-level and gene-level filtering steps, normalization, dimensionality reduction, clustering, and marker identification were performed using Seurat (v5.0.3) and Scanpy (v1.9.3), following best-practice guidelines for single-cell transcriptomic data. To ensure reproducibility and minimize batch effects, comparisons between retinal pigment epithelium and melanocyte transcriptomes were additionally conducted on samples derived from the exactly same anatomical location (anterior eye chamber). We performed multiple biological comparisons (e.g., between ocular and skin-derived melanocytes, across developmental stages, and between vertebrate and tunicate cell types) to assess the robustness of transcriptional similarities and evolutionary relationships. All single-cell clusters were annotated using well-established marker genes, and regulon analysis was performed using SCENIC with default settings and validated motif databases. All key findings were confirmed across at least two independent datasets or validated through complementary methods (e.g., transcriptomics, chromatin accessibility, and in situ hybridization). No statistical methods were used to predetermine sample sizes, as this study is based entirely on reanalysis of published datasets. No data were excluded from the analyses unless they failed established quality control thresholds. Replication was assessed computationally via cross-validation of patterns across datasets, species, and developmental stages.

### Competing interests

The authors declare no competing interests.

### Data availability

All previously published single-cell RNA-seq datasets analyzed in this study are available from the NCBI Gene Expression Omnibus (GEO) under the accession numbers: GSM5560840 (Lee et al., 2022)[33], GSE131498 (Ge et al., 2020)[43], GSE181390 (Lee et al., 2024)[44], GSE203051 (Sun et al., 2023)[45], GSE147298 (Infarinato et al., 2020)[46], GSE155121, GSE210543, and GSE131155 (Cao et al., 2019)[59]. Spatial transcriptomics data were obtained from the CNGB Nucleotide Sequence Archive under accession number CNP0001543 (Chen et al. 2022)[50].

### Code availability

All notebooks with code and processed data are deposited to GitHub: https://github.com/fateevajulia/RPE-Melanocytes_notebooks.

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

## Acknowledgements
I.A., S.I., R.G., and J.F. were supported by the ERC Synergy grant (KILL-OR-DIFFERENTIATE), Swedish Research Council, Paradifference Foundation, Cancer Foundation in Sweden, Knut and Alice Wallenberg Foundation, ALSF (the Crazy 8 project grant), Austrian Science Fund (consortium SFB F78 program and Emerging Fields "Brain Resilience" consortium, as well as stand-alone project grants). A.K. is supported by grants from the German Research Foundation (Deutsche Forschungsgemeinschaft, DFG): CRC 1461 "Neurotronics: Bio-Inspired Information Pathways" (Project-ID 434434223—SFB 1461) and KL3475/2-1. L.A.L. was supported by Provost Office of Saint Louis University (Project-ID 000487). We thank Dr. Olga Kharchenko for the help with illustrations and Dr. Alek Erickson for reading and discussing this manuscript.

## Author contributions
I.A. planted the original idea, initiated the study and supervised the work. Y.F. and R.G. performed computational data analysis and prepared the figures. S.I. supervised and advised computational analysis. A.K. assisted with phylostratigraphy analysis. L.A.L. performed all experimental work with tunicate larvae and prepared the figures. Beyond that, all authors contributed to scientific discussions, planning of work and participated in writing of the manuscript.

## Funding

## Competing interests
The authors declare no competing interests.
