## [Transparent Peer Review file · Communications Biology]

Melanocytes and photosensory organs share a common ancestry that illuminates the origins of the neural crest

Corresponding Author: Professor Igor Adameyko

This manuscript has been previously submitted at another journal. This document only contains information relating to versions considered at Communications Biology.

Version 0:

Reviewer comments:

Reviewer #1

(Remarks to the Author)

Fatieieva et al. present evidence for "deep homology" between retinal epithelial cells (RPEs) and melanocytes, despite their distinct embryonic origins from neuroepithelia and neural crest, respectively. The authors employ a variety of sophisticated computational methods, including STRING, Cassiopeia, and phylostratigraphy. The initial analyses employ previously published single cell transcriptome datasets derived from the murine eye chamber, and then extended to single cell datasets from primates. The arguments are further strengthened by the use of single cell datasets from the model tunicate, *Ciona*. This analysis provides compelling evidence that the pigmented sensory cells of the *Ciona* tadpole (the ocellus and otolith) are likely to be homologous to vertebrate RPEs and melanocytes. I think this is a significant finding.

The authors invoke this finding to propose an elaborate model for the evolutionary origins of neural crest. Much of the Discussion is highly speculative.

I suggest that the authors streamline the writing of the manuscript to emphasize the deep homology of RPEs and melanocytes. It might be interesting to compare melanocytes derived from skin and eyes. Do they really show similar homologies with RPEs, as the authors suggest?

The authors should also streamline the Discussion to emphasize the key features of their model. They should also consider alternate possibilities. For example, it's not clear that their model explains why most derivatives of neural crest correspond to neurons and pigment cells.

Reviewer #2

(Remarks to the Author)

This study re-analyzes previously published sequencing data in mice and ascidians to identify shared gene expression programs and regulatory relationships between melanocytes and retinal pigment epithelium (RPE) in vertebrates and the ocelli in ascidians, all of which combine pigmentation with photosensitivity. They find that these cell types share not only many genes and regulons associated with pigmentation, but also additional ones with distinct functions. These findings provide support for a close evolutionary relationships between these cell types and argue against the alternative hypothesis of independent recruitment of a pigmentation module into these cell types. Because melanocytes in vertebrates are neural crest (NC) derived, the authors further argue that their findings support a model of NC evolution from progenitors of pigmented photosensitive cells in the neural tube of the vertebrate tunicate ancestor.

In revealing unexpected regulatory similarities between melanocytes, RPE and ascidian ocelli, the findings of this paper give exciting new insights into the evolutionary relationships of these cell types and provide support for the idea that RPE and melanocytes may be homologous cell types, which - like the ocelli in ascidians - evolved from a common ancestral pigmented photosensitive cell type in the last common ancestor of tunicates and vertebrates. However, the presentation of this important insight is compromised by several major weaknesses of the paper in its current form, which need to be addressed before publication.

First, the paper conflates two questions, which I think should be kept separate: 1) the origin of RPE and melanocytes; and 2) the origin of the NC. The analysis presented offer interesting insights into the first question, which provides new support for the idea that RPE and melanocytes are homologous cell types. This should be the main focus and argument of the paper. If correct, such an evolutionary relationship between these cell types is also compatible with the model of NC evolution previously suggested by Ivashkin and Adameyko in 2013. However, it is also compatible with many other scenarios of NC evolution and additional evidence would be needed to specifically support the model presented. In particular, to support the idea that pigment cells played a more crucial role in NC evolution than other NC derived cells, it would be necessary to compare the transcriptomes of all NC derived cell types (including not only melanocytes but sensory neurons, glia cells, cartilage and bone, muscle cells etc.) with all cell types in *Ciona*. Most likely such an analysis will find that the most closely related cell types to these other NC derived cell types will be other *Ciona* cell types, not the ocelli, thereby providing support for the idea that the vertebrate NC is a novelty, which evolved by the cooption and recombination of GRNs from multiple cell types. Therefore, while I think it worthwhile to point out that the findings of this paper is compatible with the Ivashkin/Adameyko scenario, the sections of the paper dealing with NC origins need to be shortened and toned down. The discussion, in particular, should be rewritten to highlight the cell type homology between RPE and melanocytes as the main conclusion before a brief discussion on the potential implication for our understanding of NC origins and its limitations.

Second, I think it is not necessary to invoke “deep” homology here, which is not a conceptually very well defined notion. There has been said to be “deep homology” of structures, when the structures themselves are not homologous but some of their components, in particular parts of the gene regulatory networks building the structures are homologous. But the arguments in this paper actually do support the idea that RPE and melanocytes (and ocelli of ascidians) are truly homologous cell types, derived from a common ancestral cell type. So I strongly suggest to drop the “deep” throughout the paper and from the title.

Third, apart from these conceptual issues the discussion is rambling and poorly structured and repeats a lot of material from the results (for example, 438-489 is almost completely redundant). This should be completely rewritten and significantly shortened.

Finally, there is quite some awkward writing throughout the paper (some of which I highlight in my detailed comments below), so I recommend to let a native speaker read the paper before re-submission.

Detailed comments:

- 20: “stem chordata” is misleading here since a “stem group” refers to the extinct stem lineage of a monophyletic group outside of the “crown group” defined by its extant members
- 24/42: delete “a” before “pigmentation”
- 26: delete “a” after “revealed”
- 30/47: it should read “the vertebrate lineage”
- 34: it should read “the eye”
- 52: “from the time of cnidarians”: not clear what this means, since cnidarians are alive today
- 58: “variation of which...”: awkward sentence
- 64: it should read “the melanin...”
- 95: I agree that the NC is a novelty; but if that’s the case one should not expect to find homologues of the NC in ancestral lineages (or their descendants); while each of the cell types derived from the NC in vertebrates may have homologous cell types in such a lineages, there may be no progenitor population that can be identified as NC homologue, simply because the NC probably evolved by co-option and recombination of multiple GRNs, which were ancestrally expressed in different cells, into a single progenitor cell type
- 102: what is “developmental homology”?
- 111-118: you should be cautious about inference on neural crest origins as I argued above; it is not clear what “intercalation” and “scattering” mean here.
- 130: delete “a” before “supporting”
- 135: the concept of “deep homology” is misrepresented here. As mentioned above, there has been said to be deep homology of structures, when the structures themselves are not homologous but some of their components, in particular parts of the gene regulatory networks building the structures are homologous.
- 157: it would be helpful to briefly explain, what String analysis does
- 198: evolutionary and developmental relationships may be different; this is exactly the case for many neural crest derived cell types which have a common developmental origin but very different evolutionary relationships. For example, cartilage, muscle and neuronal cells derived from the NC all have a common developmental origin but have different closest evolutionary relationships with other (non-NC-derived) cartilage, muscle and neuronal cells, respectively.
- 207: if I understand correctly the phylogenetic tree here is based on TF expression data – please clarify
- 224: maybe write “candidate upstream TFs” for clarification
- 227 and Fig.3: I am confused, why the UMAP in Fig. 3d is not the same as in Fig. 3 a. Please explain, which clusters in Fig. 3d correspond to RPE and melanocytes
- 258: replace “where” with “with”
- 265: one problem with this analysis is, of course, that there was only a very limited number of cell types in the data set analyzed
- 269: remove “needless to say”
- 297: delete “an” before “evidence”
- 302: it should read “specialize for”

- 312-21: why is the analysis presented here relevant for the conclusions of this paper?
- 324 and following: see my comments on the discussion above: I have not attempted to improve the writing here, because this should we completely rewritten anyway.
- The figures of the main paper as well as the suppl. Figs have multiple typos and the following words are misspelt in one or multiple figs: "average", "percent", "differentially", "pericytes", "Cassiopeia", "hierarchy"
- Fig.1 : the descriptions in the legend do not match with figure panels. d and e are swapped in the legend, f is not described, g is described under "f", h and I are described under "g" and "h"
- Fig.4: d is not described, e is described under "d", f is described under "e", g is not described
- in Fig. 4 f/g a higher magnification of the eye region would be helpful
- Fig. 5; in 5b, photoreceptors should also be shown for vertebrates; in 5d Rgr and Rrh are clearly also expressed in other cell populations in Ciona, not only the ocelli; please explain in main text, which ones these are; 5j: how was the overlap between melanocytes, RPE and Ciona SV determined?
- Fig. 6, legend 828: typo "phylostratigraphy"
- Fig.7: the lower part of the figure is not described in the legend

Reviewer #3

(Remarks to the Author)

Review of the Manuscript "Deep homology of photosensory organs and melanocytes points towards an ancestral cell at the dawn of neural crest" by Yuliia Fatieieva et al.

In this study Yuliia Fatieieva et al. aims to determine whether melanocytes and retinal pigment epithelium (RPE) originate from an ancestral pigmented photosensory structure. The authors working hypothesis is that cells sharing common transcriptional programs originate from the same ancestry. To investigate this putative common origin in melanocytes and RPE, the authors gathered single-cell data from multiple studies and spatial transcriptomics data. The study combines protein network and gene ontology analyses with genomic phylostratigraphy to untangle the origin of melanocytes and RPE. First, the authors identified that RPE and melanocytes display commonly expressed genes beyond those related to pigmentation, suggesting a common ancestry. They then performed hierarchical clustering and phylogenetic tree reconstruction, highlighting the neighboring positions of RPE, melanocytes, and Schwann cells. Hierarchical clustering identifies melanocytes and Schwann cells as closest neighbors, whereas phylogenetic tree reconstruction identifies RPE and melanocytes as such. They also selected only active regulons specific to melanocytes or RPE and determined that most of them are actually shared between the two, indicating strong homology between these cell types.

In a second part, the authors investigate cells from Ciona larvae – an organism evolutionarily proximal to the split of vertebrates from chordates – focusing on pigmented cells from their sensory vesicle. Using published scRNAseq from Ciona, they reported gene expression associated with RPE and melanocytes, especially in the cluster associated with pigmented cells, and identified biological processes shared between these cell types. By performing phylogenetic tree reconstruction using transcription factors shared between rodents and tunicates, they showed that RPE and Ciona pigmented cells are neighbors, with melanocytes as the closest outgroup.

Finally, they implemented a phylostratigraphy approach to explore the evolutionary age of genes and programs specific to RPE and melanocytes, determining the genes or programs that emerged after the putative split of pigmented photoreceptors into RPE and melanocyte lineages. Based on the set of genes identified, the authors claim that functions specific to melanocytes and RPE were acquired after that split. Overall, the authors suggest that since RPE and melanocytes share common programs and function-specific genes emerged later, RPE and melanocytes are likely to share the same origin. Although the authors made efforts to investigate multiple datasets, the manuscript seems to lack strong data supporting their hypothesis. The proposed strategy is interesting, the data presented in the manuscript fall a bit short to be really convincing that melanocytes and RPE share a common cell type of origin and that evolutionary convergence is to be excluded.

Major points

1. The authors showed that RPE and melanocytes have common transcriptional programs. However, this is essentially based on melanin production ribosomes biogenesis and metabolic processes. This evidence seems insufficient to favor the common ancestry hypothesis over the convergent evolution one. To support their claim, the authors need to compare – if such exist – cell types known to have a common ancestry as a reference point.

2. The study focuses on the eye and compares RPE and melanocytes from the same organ. The fact that these cells are localized in the same region could also explain why they share transcriptional programs, as they are exposed to the same surrounding signals and environment. The authors need to add data from skin melanocytes in their analysis to prove this similarity in transcriptional programs is really reflecting a putative common origin or if on the contrary it reflects an evolutionary convergence linked to the cell being positioned in the same organ. Identifying common programs among these three cell types would support the claim better than just comparing eye cell types.

3. The authors based their analyses solely on gene expression datasets. Changes in sequences of distal regulatory elements has been showed to be a major driver of evolution as it can happen much more frequently than mutation in transcription factor binding or changes in gene function. In the present study, chromatin accessibility and regulatory elements have not been studied. Comparing and analyzing such datasets could enrich the study, as revealing common processes at this level would provide additional evidence for a shared origin.

4. The authors used the Ciona larva as an outgroup. They highlighted a set of markers for melanocytes and RPE in Ciona larva, particularly in the pigmented photoreceptor cluster. Images showing the expression of these markers in the larva would be interesting to estimate the specificity of these markers within the organism and identify other cell types expressing them. This is essential since there are many examples in the literature showing that ancestral genes in amphioxus or

lamprey are expressed during development but either at different timing or location or have even have different functions.

5. The manuscript would greatly benefit from a detailed description (rather than an overview) of all figures, as well as thorough proofreading of the abstract, text, and figure legends. Several passages are extremely deterministic with conclusion drawn without evidences. Finally, several paragraphs needs to be rewritten in a more scientific style. For example, a sentence like "Needless to say that » are really not welcome in a scientific manuscript.

Version 1:

Reviewer comments:

Reviewer #1

(Remarks to the Author)

The authors have significantly improved the evidence for deep homology of RPEs and melanocytes. The new results presented in Fig. 5 are a nice addition to the revised manuscript. My only lingering concern is the length of the Discussion. It is currently 5 pages long and could easily be trimmed to 3 pages by eliminating repetitions with the Intro and Results section.

Reviewer #2

(Remarks to the Author)

In their revised manuscript Fatieieva et al. carefully addressed y comments and those of the other reviewers. They added substantial new data analyses and completely rewrote the manuscript, which is now significantly improved.

I have only a few minor comments remaining:

- 123-126: I think homology is about identity rather than similarity and would thus prefer to word this differently (although I accept that the authors may disagree), such as : "Homology refers to the identity of structures in different species due to common ancestry, which may not be immediately apparent in distantly related species, where structures may have become dissimilar due to evolutionary convergence" and provide a different reference (e.g . Wagner, G. P. 2014. Homology, Genes, and Evolutionary Innovation. Princeton: Princeton University Press)
- 470-473: This sentence confuses rather than clarifies what the authors propose. In the next paragraph they state that skeletogenic fates may have been coopted secondarily into the NC lineage. So it is confusing that they write here "and beyond, as seen in the case of mesoderm-like ectomesenchyme". I also find it misleading to talk about "functionally analogous" cell types here, since their central argument seems to be about homologous cell types. Thus I suggest to simplify and reword the sentence as follows: "This idea is supported by the observation that the neural crest frequently generates cell types that are identical to those produced by other progenitor populations within the CNS."
- 507: delete "in lineage"
- 517: add "we" before performed"
- There are still a few typos in the figures or figure legends (e.g. for Cassiopeia in Fig. 3c, Suppl. Figs 2 and 6); for pericyte in Fig. 1a, for melanocyte in Fig. 7a);
- abbreviation "ov" in Fig. 5d needs to be explained in legend;
- the lower part of Suppl. Fig. 4d is not explained in legend

Reviewer #3

(Remarks to the Author)

The authors addressed all my remarks in the revised version of this manuscript which now is much clearer and convincing. It is now ready for publication

Some minor comments below that need to be corrected in the text

1. Line 160: "Although these genes might be stronger enriched "
2. Line 162 – 163: "Gene ontology enrichment and STRING analysis of these common melanocyte (MEL) + RPE genes revealed that"
3. Line 516-517: The term "pools of DNA Controllers » is very unclear. Authors should use transcription factor or DNA binding proteins instead
4. Line 523: When using the term "transcriptional controllers" are the authors referring to mechanisms regulating the rate of transcription or to DNA binding proteins? The authors need to clarify this sentence
5. Line 533: The authors need to replace the term "regulatory controllers" by something more generally used by the community
6. Fig.2c: $-\log_{10}(\text{FDR})$ scale value is missing
7. Fig. 5d: It is indicated Melano but authors should write Melanocytes

Response to reviewer's comments

Reviewer #1 (Remarks to the Author):

Fatieieva et al. present evidence for "deep homology" between retinal epithelial cells (RPEs)
and melanocytes, despite their distinct embryonic origins from neuroepithelia and neural crest,
respectively. The authors employ a variety of sophisticated computational methods, including
STRING, Cassiopeia, and phylostratigraphy. The initial analyses employ previously published
single cell transcriptome datasets derived from the murine eye chamber, and then extended to
single cell datasets from primates. The arguments are further strengthened by the use of
single cell datasets from the model tunicate, *Ciona*.

This analysis provides compelling evidence that the pigmented sensory cells of the *Ciona*
tadpole (the ocellus and otolith) are likely to be homologous to vertebrate RPEs and
melanocytes. I think this is a significant finding.

- We thank the reviewer for positive take on our study, and we hope that the revised
manuscript provides even more compelling evidence in support of homology of RPE,
melanocytes and *Ciona* pigmented photosensory cells. We sincerely appreciated the
reviewer's insightful comments. We acknowledge the importance of refining our
manuscript to better streamline our discussion, provide stronger evidence, and
consider alternative interpretations of our results. Below, we provide detailed
responses to each of the key points raised by the reviewer.

The authors invoke this finding to propose an elaborate model for the evolutionary origins of
neural crest. Much of the Discussion is highly speculative. I suggest that the authors
streamline the writing of the manuscript to emphasize the deep homology of RPEs and
melanocytes.

- We agree and we reorganized the manuscript by adding new analysis with tighter focus
on homology of different pigmented photosensory cells from mouse and tunicate. For
instance, we have added new analyses comparing chromatin profiles in RPE,
melanocytes, and other cell types (see **new Supplementary Figures 4 and 5**).
Additionally, we have expanded our comparative framework by incorporating skin
melanocytes from several published studies (see **new Supplementary Figure 2**).
Additionally, we experimentally validated the expression of key genes in *Ciona* larvae
by using HCR in situ hybridization (**updated Figure 5e**). We have also streamlined the
Discussion, reducing the section dedicated to the neural crest evolution scenario.
While we could not entirely omit this aspect - that our study holds relevance in the
context of the neural crest's evolutionary origin - we have made an effort to present a
more balanced and focused narrative. We tried to balance the reorganization of the
text, addition of new principal results and requested (by other reviewer) detalization of
Results. Overall, we hope we achieved a new balance of details and the general
narrative, which will hopefully make the reviewers happy.

It might be interesting to compare melanocytes derived from skin and eyes. Do they really
show similar homologies with RPEs, as the authors suggest?

- We fully agree with this suggestion and we did as the reviewer proposed. We
performed the comparison of RPE and skin melanocytes in a context of other cell types
by merging anterior eye chamber dataset with RPE cells with four different skin
melanocyte clusters from other published datasets (injecting different skin melanocyte
clusters into comparison one by one): postnatal day 0 (GSE131498, mouse back
skin), postnatal day 4 (GSE181390, mouse back skin), five-week-old mice
(GSE203051, activated melanocyte stem cells from anagen II hair follicles) and
quiescent melanocyte stem cells (GSE147298, adult hair follicles). All datasets were
processed using Scanpy (v1.9.1) in Python (v3.8.10), applying rigorous quality control,
normalization, batch correction (Harmony v0.1.0), clustering (Leiden algorithm), and
visualization (UMAP), differential gene expression analysis. Hierarchical clustering
nested the RPE cluster most proximally to melanocytes from both eye and skin, as
compared to other cell types. Most importantly, CASSIOPEIA-based parsimony
analysis based on transcription factors consistently grouped RPEs with skin-derived
melanocytes across all datasets. Skin-derived melanocytes turned out to be more
similar to RPE as compared to eye-derived melanocytes in Cassiopeia analysis. This
recurrent clustering pattern suggests that melanocytes from different spatial origins
retain core transcriptional features that link them to RPEs in terms of transcriptional
similarity (**see new Supplementary Figure 2**). Noteworthy, we performed our initial
comparisons between ocular melanocytes and RPE because they come from the same
dataset, which guarantees no batch effects and less noisy analysis. Batch effects in
single cell data can be detrimental and potentially may cause misleading results. Now,
we can see that possible batch effects are not strong enough to perturb grouping of
different skin and eye melanocytes with RPE, independently of their dataset and body
location origin. Finally, we rigorously tested the parsimony-based trees obtained with
CASSIOPEIA tool based on presence of DNA-binding proteins. For this, we added
more cell types into the comparison also using those, for which evolutionary
relationships are known: astrocytes and Muller glia, rods and cones, immune cells. The
new results supported the adequacy of the chosen methodology (see **new**
**Supplementary Figure 6**) We provided these details and explanations in the **updated**
**Results** section.

The authors should also streamline the Discussion to emphasize the key features of their
model. They should also consider alternate possibilities. For example, it's not clear that
their model explains why most derivatives of neural crest correspond to neurons and
pigment cells.

- We agree, and we presented the debates about the neural crest evolution in a clearer
and streamlined way. We also mentioned possible alternatives and ensured a clearer
transition in the evolutionary logic. We added this paragraph to the revised Discussion
in response to this comment:

*“What, then, about the other neural crest-derived cell types within this evolutionary*
*framework? It is plausible that the progenitors giving rise to melanocytes were already*
*multipotent at the time of their emigration from the neural tube (Todorov et al., 2024),*
*allowing them to generate not only pigment cells but also other neural crest derivatives*

such as peripheral sensory neurons and glia. This idea is supported by the observation
that the neural crest frequently generates cell types that are functionally analogous -
or even identical - to those produced by other progenitor populations both within the
CNS and beyond, as seen in the case of mesoderm-like ectomesenchyme. A well-
known example of such functional duplication involves Rohon-Beard neurons,
somatosensory neurons that remain within the spinal cord, extend axons into the
periphery, and exhibit diverse sensory modalities. These neurons co-exist alongside
neural crest-derived dorsal root ganglion (DRG) neurons, which fulfill comparable
sensory roles, and this overlap can persist for extended developmental periods, if not
indefinitely (Donoghue et al., 2008; Liu and Kucenas, 2024). This redundancy suggests
that the emergence of neural crest derivatives was not strictly about inventing novel
cell types, but rather about repositioning existing functions (including light-responding
pigmentation) into new anatomical and ecological contexts - outside the CNS - through
the mobilization of multipotent progenitors.

A central question in this context is what evolutionary and ecological pressures drove
the emergence of the neural crest, and which of its diverse derivatives represent
ancestral functions versus later co-opted fates. Our core hypothesis posits that vision
and pigmentation played a foundational role in the origin of the neural crest. We argue
that these functions - particularly through light-responsive pigment cells - offered early
selective advantages that could have favored the evolution of migratory, multipotent
progenitors.

In contrast, alternative candidate cell types such as sensory neurons or glia
appear less likely to have served as primary drivers. This is supported by the existence
of Rohon-Beard neurons, which fulfill peripheral sensory functions while remaining
within the CNS, as well as the mesodermal properties of ectomesenchymal cells, which
suggest a later integration into the neural crest lineage. Thus, the emergence of neural
crest identity may have initially been rooted in light-sensitive pigment cells, with other
fates incorporated over time through evolutionary co-option of cell fates and sub-
lineages instead of separate function-related gene expression programs”

Also, we mention the main alternative already in the Results section:

“This result rather supports that RPE and melanocytes might share the evolutionary
origin, and argue against an alternative - a simple co-option of a pigmentation gene
expression circuit into some enigmatic cell type in basal chordates.”

and

“Further gene expression analysis identified a subset of genes commonly expressed
in both melanocytes and RPE cells (Fig. 2a). This is relevant for testing whether
cooption of a specific gene circuit occurred, or whether diverse classes of gene
functions were shared, which would instead support homology (Fig. 2b).”

**Reviewer #2 (Remarks to the Author):**

This study re-analyzes previously published sequencing data in mice and ascidians to identify
shared gene expression programs and regulatory relationships between melanocytes and
retinal pigment epithelium (RPE) in vertebrates and the ocelli in ascidians, all of which combine
pigmentation with photosensitivity. They find that these cell types share not only many genes
and regulons associated with pigmentation, but also additional ones with distinct functions.
These findings provide support for a close evolutionary relationships between these cell types
and argue against the alternative hypothesis of independent recruitment of a pigmentation
module into these cell types. Because melanocytes in vertebrates are neural crest (NC)
derived, the authors further argue that their findings support a model of NC evolution from
progenitors of pigmented photosensitive cells in the neural tube of the vertebrate tunicate
ancestor. In revealing unexpected regulatory similarities between melanocytes, RPE and
ascidian ocelli, the findings of this paper give exciting new insights into the evolutionary
relationships of these cell types and provide support for the idea that RPE and melanocytes
may be homologous cell types, which - like the ocelli in ascidians - evolved from a common
ancestral pigmented photosensitive cell type in the last common ancestor of tunicates and
vertebrates. However, the presentation of this important insight is compromised by several
major weaknesses of the paper in its current form, which need to be addressed before
publication.

- We truly appreciate the reviewer's constructive feedback and recognition of the
significance of the evolutionary relationships between melanocytes, retinal pigment
epithelium (RPE), and ascidian pigmented ocelli. Below, we've provided detailed
responses to each concern and outlined the improvements. Thank you again for your
valuable insights - we genuinely appreciate the time and effort put into these
comments.

First, the paper conflates two questions, which I think should be kept separate: 1) the origin of
RPE and melanocytes; and 2) the origin of the NC. The analysis presented offers interesting
insights into the first question, which provides new support for the idea that RPE and
melanocytes are homologous cell types. This should be the main focus and argument of the
paper. If correct, such an evolutionary relationship between these cell types is also compatible
with the model of NC evolution previously suggested by Ivashkin and Adameyko in 2013.

- In response to this comment, we improved the narrative structure by first discussing
these two questions separately and then connecting them more effectively in the
Introduction and Discussion. For us, the origin of the neural crest remains the central
question, and we approach the homology of RPE and melanocytes as a means to
address this broader research problem. Our study of RPE and melanocyte
transcriptional programs was specifically driven by the question related to neural crest
evolution, not the other way around. We believe that the homology between RPE and
melanocytes holds limited interest outside the context of neural crest research, and we
do not know how to explain why we decided to study this homology if not because of
the neural crest origin problem. Therefore, we organized the revised flow of the
manuscript allowing readers to follow our line of reasoning. Altering this structure would
result in a narrative that feels detached and lacks a clear purpose, as it would obscure
the fundamental motivation behind our research.

In our view, the two questions mentioned by the reviewer are inherently linked, yet we
recognize the need to present our reasoning more clearly and systematically. To
address this, we added new passages to the Discussion to better illustrate how these
questions are interrelated and why the homology between RPE and melanocytes
becomes particularly significant when considered within the framework of neural crest
origin. We believe that the revised manuscript now presents a more coherent
connection, eliminating any sense of disjointedness, particularly in the section related
to neural crest evolution. This restructuring was also motivated by Reviewer 1, who
requested a clearer explanation of the evolutionary alternatives.

Importantly, during this revision, we focused on expanding RPE-melanocytes part by
adding new analysis addressing the similarity of RPE and melanocytes at the level of
chromatin accessibility landscapes and by adding more cell types and datasets into
comparisons of transcriptional states (see **new Supplementary Figures 2,4,5,6**). We
hope that this amount of new information shifts the balance of discussion towards cell
type homology (RPE, melanocytes and *Ciona* sensory vesicle), as reviewer advised.

However, it is also compatible with many other scenarios of NC evolution and additional
evidence would be needed to specifically support the model presented. In particular, to support
the idea that pigment cells played a more crucial role in NC evolution than other NC derived
cells, it would be necessary to compare the transcriptomes of all NC derived cell types
(including not only melanocytes but sensory neurons, glia cells, cartilage and bone, muscle
cells etc.) with all cell types in *Ciona*. Most likely such an analysis will find that the most closely
related cell types to these other NC derived cell types will be other *Ciona* cell types, not the
ocelli, thereby providing support for the idea that the vertebrate NC is a novelty, which evolved
by the cooption and recombination of GRNs from multiple cell types.

- We agree with these doubts and we performed **additional investigations** by adding
more cell types into the comparisons of transcriptional profiles (see **new**
**Supplementary Figure 6**) and also additionally investigated the regulatory
landscapes including comparative chromatin accessibility assays (**new**
**Supplementary Figures 4 and 5**). The obtained results supported the homology of
RPE and melanocytes, which is the main line of investigation and the foundational
support for our evolutionary scenario of the neural crest origin. **Importantly, to**
**address this comment further**, we provided the following explanation (Discussion
section) for the problem of alternative cell types being more homologous to ancient
neural crest cells and their derivatives, as compared to pigment photoreceptor cell
types:

*“What, then, about the other neural crest-derived cell types within this evolutionary*
*framework? It is plausible that the progenitors giving rise to melanocytes were already*
*multipotent at the time of their emigration from the neural tube (Todorov et al., 2024),*
*allowing them to generate not only pigment cells but also other neural crest derivatives*
*such as peripheral sensory neurons and glia. This idea is supported by the observation*
*that the neural crest frequently generates cell types that are functionally analogous -*
*or even identical - to those produced by other progenitor populations both within the*
*CNS and beyond, as seen in the case of mesoderm-like ectomesenchyme. A well-*
*known example of such functional duplication involves Rohon-Beard neurons,*
*somatosensory neurons that remain within the spinal cord, extend axons into the*

*periphery, and exhibit diverse sensory modalities. These neurons co-exist alongside*
*neural crest-derived dorsal root ganglion (DRG) neurons, which fulfill comparable*
*sensory roles, and this overlap can persist for extended developmental periods, if not*
*indefinitely (Donoghue et al., 2008; Liu and Kucenas, 2024). This redundancy suggests*
*that the emergence of neural crest derivatives was not strictly about inventing novel*
*cell types, but rather about repositioning existing functions (including light-responding*
*pigmentation) into new anatomical and ecological contexts - outside the CNS - through*
*the mobilization of multipotent progenitors.*

*A central question in this context is what evolutionary and ecological pressures drove*
*the emergence of the neural crest, and which of its diverse derivatives represent*
*ancestral functions versus later co-opted fates. Our core hypothesis posits that vision*
*and pigmentation played a foundational role in the origin of the neural crest. We argue*
*that these functions - particularly through light-responsive pigment cells - offered early*
*selective advantages that could have favored the evolution of migratory, multipotent*
*progenitors.*

*In contrast, alternative candidate cell types such as sensory neurons or glia*
*appear less likely to have served as primary drivers. This is supported by the existence*
*of Rohon-Beard neurons, which fulfill peripheral sensory functions while remaining*
*within the CNS, as well as the mesodermal properties of ectomesenchymal cells, which*
*suggest a later integration into the neural crest lineage. Thus, the emergence of neural*
*crest identity may have initially been rooted in light-sensitive pigment cells, with other*
*fates incorporated over time through evolutionary co-option of cell fates and sub-*
*lineages instead of separate function-related gene expression programs”*

Therefore, while I think it worthwhile to point out that the findings of this paper is compatible
with the Ivashkin/Adameyko scenario, the sections of the paper dealing with NC origins need
to be shortened and toned down. The discussion, in particular, should be rewritten to highlight
the cell type homology between RPE and melanocytes as the main conclusion before a brief
discussion on the potential implication for our understanding of NC origins and its limitations.

- - As we mentioned above, we have naturally expanded the sections related to the
homology of RPE and melanocytes due to the inclusion of new results during this
revision. For example, we have conducted new analyses comparing chromatin profiles
in RPE, melanocytes, and other cell types (see **new Supplementary Figures 4 and**
**5**). We have also expanded our comparative transcriptomics framework by including
skin melanocytes and additional retinal cell types from several published studies (see
**new Supplementary Figures 2 and 6**). Furthermore, we experimentally validated the
expression of key genes in *Ciona* larvae using HCR *in situ* hybridization (**updated**
**Figure 5e**). In the final part of the revised manuscript, we expanded into the discussion
of the inherent limitations of using modern methods to investigate the homology of RPE
and melanocytes. Together, this significantly shifted the balance of discussion towards
RPE and melanocytes homology.

Second, I think it is not necessary to invoke “deep” homology here, which is not a conceptually
very well defined notion. There has been said to be “deep homology” of structures, when the
structures themselves are not homologous but some of their components, in particular parts
of the gene regulatory networks building the structures are homologous. But the arguments in

this paper actually do support the idea that RPE and melanocytes (and ocelli of ascidians) are
truly homologous cell types, derived from a common ancestral cell type. So I strongly suggest
to drop the “deep” throughout the paper and from the title.

- We fully agree with the reviewer on this matter and we decided to abandon the “deep”
homology concept. This is fully corrected now.

Third, apart from these conceptual issues the discussion is rambling and poorly structured and
repeats a lot of material from the results (for example, 438-489 is almost completely
redundant). This should be completely rewritten and significantly shortened.

- We rewrote the discussion in more structured way, and we hope that the reviewer will
feel the difference.

Finally, there is quite some awkward writing throughout the paper (some of which I highlight
in my detailed comments below), so I recommend to let a native speaker read the paper before
re-submission.

- We asked a native speaker to proofread the paper (our American postdoc). We hope
there are no more awkward phrases left in the text. We thank the reviewer for helping
295 us to improve the style and the content.

***Detailed comments:***

300 - 20: “stem chordata” is misleading here since a “stem group” refers to the extinct stem lineage
of a monophyletic group outside of the “crown group” defined by its extant members.

- We agree, and we removed this term.

305 - 24/42: delete “a” before “pigmentation”

- Corrected

309 - 26: delete “a” after “revealed”

- Done

313 - 30/47: it should read “the vertebrate lineage”

- Corrected

317 - 34: it should read “the eye”

- Done

321 - 52: "from the time of cnidarians": not clear what this means, since cnidarians are alive
today

- We clarified the intended meaning, thanks for spotting this. Now the phrase reads:
"*Melanin and photoreception have been closely linked since the emergence of*
*cnidarians, as both are present in cnidarian sensory organs, including the camera eyes*
*of cubozoan jellyfish*"

329 - 58: "variation of which...": awkward sentence

- We corrected the sentence: "*Even in vertebrates, the third eye, also known as a*
*pineal gland, contains melanin-bearing pinealocytes*"

334 - 64: it should read "the melanin..."

- Corrected

338 - 95: I agree that the NC is a novelty; but if that's the case one should not expect to find
homologues of the NC in ancestral lineages (or their descendants); while each of the cell types
derived from the NC in vertebrates may have homologous cell types in such a lineages, there
may be no progenitor population that can be identified as NC homologue, simply because the
NC probably evolved by co-option and recombination of multiple GRNs, which were
ancestrally expressed in different cells, into a single progenitor cell type.

- Although we agree with the reviewer's logic, it is important to recognize that
evolutionary novelty does not arise without a trace or origin. Novel traits inevitably
emerge from pre-existing evolutionary substrates through processes such as
modification, co-option, and recombination. In this study, we aim to identify which
ancient cell identities were initially recruited and subsequently modified to give rise to
the entire neural crest lineage (and we do not discuss much the homologous
embryonic progenitors of delaminating and migratory neural crest). Our hypothesis is
that pigmented photoreceptors and their immediate progenitors, located within the
chordate central nervous system, represent a plausible substrate that underwent
evolutionary changes, ultimately leading to the emergence of the neural crest lineage
as a fundamental novelty. Once this cell lineage acquired features that conferred
evolutionary advantages associated with the neural crest, it likely became integrated
as a highly distinct embryonic cell lineage. Again, it is important to clarify that our focus
is not on identifying embryonic homologs of the delaminating and migrating neural
crest (which would likely be neuroprogenitors resembling radial glia cells). Instead, we
propose a conceptual framework for understanding the evolutionary drivers behind
neural crest emergence, with particular emphasis on how pigment cells may have
played a pivotal role in this process.

In sum, we align with the reviewer's logic, as it does not contradict our interpretation
presented in the manuscript. We have revised the Discussion section to more clearly
articulate this perspective, particularly emphasizing the roles of pigmentation and
photoreception as key evolutionary drivers leading to re-positioning of CNS progenitors
to body periphery via emigration in ancient chordates.

- 102: what is “developmental homology”?

- We re-explained this part in different words: *“Because the embryological origin of RPE and pinealocytes is different to a significant extent from the neural crest and neural crest-derived melanocytes, such disconnection in terms of developmental origin is a major factor which precludes homologizing RPE cells, pinealocytes and scattered melanocytes. However, is there such an unbridgeable divide, and is it worth revisiting the homology of these cell types in light of the most recent knowledge?”*

- 111-118: you should be cautious about inference on neural crest origins as I argued above; it is not clear what “intercalation” and “scattering” mean here.

- We clarified the relevant text: *“Apparently, both neural crest cells, which give rise to melanocytes, and pigmented photosensory organs originate from embryonic neuroectodermal tissue, although they emerge from distinct, differentially patterned sub-regions. However, neural crest cells (along with their intermediate multipotent derivatives, such as Schwann cell precursors) have evolved as an intermediary developmental stage between the embryonic neuroepithelium and the terminal photosensory pigmented melanocytes. This intermediate step likely facilitates the widespread dispersal of melanogenic progenitors throughout the body, including the skin.”*

- 130: delete “a” before “supporting”

- Done

- 135: the concept of “deep homology” is misrepresented here. As mentioned above, there has been said to be deep homology of structures, when the structures themselves are not homologous but some of their components, in particular parts of the gene regulatory networks building the structures are homologous.

- We fully agree and we got rid of this concept throughout the manuscript.

- 157: it would be helpful to briefly explain, what String analysis does

- We added an explanation of STRING analysis in the Methods section: *“The STRING (Search Tool for the Retrieval of Interacting Genes/Proteins) tool is an online resource used to analyze protein-protein interactions and functional associations from gene lists. It integrates information from various sources, including experimental data, computational predictions, and public text collections, to build interaction networks.”*
In brief, STRING tool reveals enriched or overrepresented interactions between proteins – it creates a graph reflecting functional protein interactions in cells. We would like to emphasize that it is backed by a wealth of empirical data. In our case, STRING highlighted protein networks beyond pigmentation, which are shared by RPE and melanocytes, including metabolic and photosensory pathways.

414 - 198: evolutionary and developmental relationships may be different; this is exactly the case
for many neural crest derived cell types which have a common developmental origin but very
different evolutionary relationships. For example, cartilage, muscle and neuronal cells derived
from the NC all have a common developmental origin but have different closest evolutionary
relationships with other (non-NC-derived) cartilage, muscle and neuronal cells, respectively.

- We appreciate the reviewer's comment and agree with the point raised. In some cases,
transcriptional similarity arising from a shared developmental origin is both detectable
and pronounced. A prime example of this is the relationship between melanocytes and
Schwann cell precursors/Schwann cells. Melanocytes emerge from the Schwann cell
lineage during development (Adameyko et al., Cell 2009), and even in their fully mature
state, these cell types exhibit significant transcriptional similarity, including the
expression of key genes such as the transcription factor *Sox10*. This similarity is well
understood within the framework of developmental biology, as it reflects a closely
related developmental trajectory and the inheritance of shared transcriptional
programs.

However, the situation differs in the case of melanocytes and retinal pigment
epithelium (RPE) cells. These two cell types do not share a proximally common
developmental origin, yet they exhibit similarities in their transcriptional programs. In
this context, the observed transcriptional resemblance is not derived from a shared
developmental pathway but rather from evolutionary convergence or homology of gene
expression programs and cell identities. The analytical approach we employ does not
inherently distinguish between similarities due to common developmental origin and
those resulting from evolutionary convergence of transcriptional programs.
Nevertheless, this does not pose a conceptual problem in our study, as it is well-
established that melanocytes and RPE cells are developmentally distinct in
vertebrates. Therefore, we interpret their similarities as reflecting evolutionary, rather
than developmental, relationships. We discussed this in the revised manuscript:

*“After constructing a dendrogram based on hierarchical clustering of global*
*transcriptional profiles (considering all genes), we observed that RPE cells,*
*melanocytes from both ocular and skin origins, and Schwann cells consistently*
*clustered in close proximity (Supplementary Fig. 2a-c, e-g, i-k, m-o). The close*
*association of Schwann cells is biologically anticipated, as melanocytes are known to*
*arise developmentally from the Schwann cell lineage (Adameyko et al., 2012, 2009).*
*Such persistence of transcriptional similarity reflects their immediate shared*
*developmental history, which lends additional confidence to the robustness and*
*biological relevance of our analysis, demonstrating that developmentally defined*
*relationships can be faithfully recovered through unbiased transcriptome-wide*
*comparisons.”*

456 - 207: if I understand correctly the phylogenetic tree here is based on TF expression data –
457 please clarify.

- Yes, you are absolutely correct, it is based on TF expression data. We made sure it is
clear in the main text: *“Using parsimony, we can directly compare the sets of expressed*
*transcription factors, where every TF is either present or absent (Joshi and Göttgens,*

2011). We generated such sets for every cluster by filtering the single cell data applying
stringent criteria (see Methods section for details).”

465 - 224: maybe write “candidate upstream TFs” for clarification

- We agree. This suggestion is implemented now.

469 - 227 and Fig.3: I am confused, why the UMAP in Fig. 3d is not the same as in Fig. 3 a.

Please explain which clusters in Fig. 3d correspond to RPE and melanocytes

- Thank you for this comment. We improved Figure 3 by showing “annotated cluster
embeddings” for each case – TF-based embedding in a-c and all genes-based
embedding in d. Just to additionally clarify: **Figure panels 3a, 3b, and 3c** show an
exclusively transcription factor-based analysis, UMAP visualization, hierarchical
clustering, and the CASSIOPEIA parsimony-based analysis, whereas embeddings in
**Figure 3d–e** are based on regulon activity and overall gene expression (same
embedding as in **Figure 1b**). Cluster annotations in **Figure 3d** correspond to those in
**Figure 1b**, with RPE and melanocyte clusters clearly labeled. Now, we improved the
figure and show both types of annotated embeddings to avoid any misinterpretation.

483 - 258: replace “where” with “with”

- Done

487 - 265: one problem with this analysis is, of course, that there was only a very limited number
of cell types in the data set analyzed.

- We agree. To respond to this comment, we conflated the original dataset of the anterior
eye chamber with the posterior eye chamber dataset (using the single cell atlas
published by Jin Li et al., 2024), to significantly increase the number of analyzed cell
types. Also, we selected this retinal dataset for checking if our analysis (Cassiopeia
and other approaches) recovers the known ground truth – the evolutionary
relationships between rods and cones, as well as bipolar cells, RGS cells, glial cell
types, immune cell subtypes etc. The results of this expanded analysis confirmed the
previous observations and also additionally supported the choice of our methodology
– please see new Supplementary Figure 3.

500 - 269: remove “needless to say”

- Done

504 - 297: delete “an” before “evidence”

- Done

508 - 302: it should read “specialize for”

- Corrected
512 - 312-21: why is the analysis presented here relevant for the conclusions of this paper?
- This specific part of the phylostratigraphy analysis aimed at balancing the overall
picture, from the position of “recruitment of old genes and programs” to “inventing new”,
therefore providing a more holistic, comprehensive view. As a part of this perspective,
we intended to uncover evolutionary young genes, which independently incorporated
into both RPE and melanocytes for some reason. Such young genes were retained in
the analysis only if they emerged after 10th phylostratum. After performing this analysis,
we found the universal incorporation of evolutionary young immunity and immune
system-interacting programs into both cell types, coinciding with major landmarks of
immune system evolution in vertebrates (integration of MHCs etc.). We attempted to
improve the clarity of the discussion of these results in the revised version of the
manuscript. Overall, this result highlighted the common lines in tissue integration
programs of melanocytes and RPE during evolution of vertebrates and after the initial
divergence of these cell identities.
528 - 324 and following: see my comments on the discussion above: I have not attempted to
529 improve the writing here, because this should we completely rewritten anyway.
- We payed special attention to updating the Discussion - lines (324+). We hope the
logical flow and connection between different parts significantly improved.
- The figures of the main paper as well as the suppl. Figs have multiple typos and the following
words are misspelt in one or multiple figs: “average”, “percent”, “differentially”, “pericytes”,
“Cassiopeia”, “hierarchy”
- Thank you for spotting these errors. We have carefully reviewed all figures in both
the main manuscript and the supplementary materials, and corrected the misspelling.
- Fig.1: the descriptions in the legend do not match with figure panels. d and e are swapped
in the legend, f is not described, g is described under “f”, h and l are described under “g” and
“h”
- This is now corrected.
- Fig.4: d is not described, e is described under “d”, f is described under “e”, g is not
described
- Corrected
- in Fig. 4 f/g a higher magnification of the eye region would be helpful
- We have updated Fig. 4f/g to include a more zoomed-in version of the cranial region.
The eye is clearly visible now, plus we introduced the arrows and the dotted lines
showing the relevant area for comparisons.

- Fig. 5; in 5b, photoreceptors should also be shown for vertebrates;

- We agree, though it would take too much space in Figure 5, so we devised the entirely **new Supplementary Figure 6**, where we show the expression of key cell types and genes in mouse photosensory retina (using the dataset from Jin Li et al., 2024) and also reveal the parsimony-based tree of multiple cell types including photoreceptors of retina (rods, cones), RPE and melanocytes. We hope this new analysis perfectly answers the reviewer's request.

in 5d Rgr and Rrh are clearly also expressed in other cell populations in *Ciona*, not only the ocelli; please explain in main text, which ones these are;

- We thank the reviewer for this comment. Indeed, *Rrh* and other genes are expressed in other parts of the developing *Ciona* nervous system proximal to sensory vesicle, which we experimentally validated by HCR *in situ* hybridization (for *Rrh*, see **revised Figure 5**). For instance, *Rrh* was found to be expressed in neuroprogenitors and ependymal cells in addition to pigmented ocellus according to HCR and single cell transcriptomics. As requested, we now clarify in the main text that *Rrh* and other markers are also expressed in other *Ciona* nervous system cell populations beyond the ocellus cluster: "*The expression patterns of Otx, Rpe65 and Rrh revealed a striking co-localization with Tyr, in the pigmented cells of Ciona larvae (Fig. 5e). Moreover, Hh2 expression overlap with this domain, consistent with some of these ependymal neural progenitors sharing a common developmental lineage with the pigment cells (Todorov et al., 2024).*"

There is a hypothesis (currently unpublished) that the larvae of aquatic animals pace their CNS maturation via light detection in neuroprogenitors – to better control circadian and lunar periodicity and slow down or speed up their development depending on the time of the year and food availability in a plankton. As far as we are aware of, parallel efforts of another lab (our colleagues) address specifically this question in more detail. Hence, we prefer not to reveal more information because of ethical concerns.

5j: how was the overlap between melanocytes, RPE and *Ciona* SV determined?

- In the revised manuscript, we decided to remove this visualization, as it does not add more to the story after additional experiments, and the exact number of overlapping genes depends on data filtration criteria, which are described in methods of the initial submission. We think it might be counterproductive and even misleading to give a precise number of overlapping genes because this number can fluctuate to some extent depending on how we filter genes.
Regarding how that number was calculated in the previous version: we identified genes significantly expressed in $\geq 10\%$ of individual cells per cluster (melanocytes, RPE, and *Ciona* sensory vesicle), mapped *Ciona* genes to mouse orthologs, and used set-based intersection analysis to calculate shared genes. This approach was detailed in the Methods and visualized using a Venn diagram (which was fully explained at the level of the code in Notebook 5f_1). If you wish to further check how that was implemented

in the actual code, please navigate to the initial submission files in the online system,
which you shall be able to access. Currently, all of that is removed from the current
resubmission.

- Fig. 6, legend 828: typo “phylostratigraphy”

- We fixed the typo, thanks a lot!

- Fig.7: the lower part of the figure is not described in the legend

- We are sorry for this error. We added the description of the lower part of the Fig.7.

**Reviewer #3 (Remarks to the Author):**

Review of the Manuscript “Deep homology of photosensory organs and melanocytes points
towards an ancestral cell at the dawn of neural crest” by Yuliia Fatieieva et al. In this study
Yuliia Fatieieva et al. aims to determine whether melanocytes and retinal pigment epithelium
(RPE) originate from an ancestral pigmented photosensory structure. The authors working
hypothesis is that cells sharing common transcriptional programs originate from the same
ancestry. To investigate this putative common origin in melanocytes and RPE, the authors
gathered single-cell data from multiple studies and spatial transcriptomics data. The study
combines protein network and gene ontology analyses with genomic phylostratigraphy to
untangle the origin of melanocytes and RPE.

First, the authors identified that RPE and melanocytes display commonly expressed genes
beyond those related to pigmentation, suggesting a common ancestry. They then performed
hierarchical clustering and phylogenetic tree reconstruction, highlighting the neighboring
positions of RPE, melanocytes, and Schwann cells. Hierarchical clustering identifies
melanocytes and Schwann cells as closest neighbors, whereas phylogenetic tree
reconstruction identifies RPE and melanocytes as such. They also selected only active
regulons specific to melanocytes or RPE and determined that most of them are actually shared
between the two, indicating strong homology between these cell types.

In a second part, the authors investigate cells from *Ciona* larvae – an organism evolutionarily
proximal to the split of vertebrates from chordates – focusing on pigmented cells from their
sensory vesicle. Using published scRNAseq from *Ciona*, they reported gene expression
associated with RPE and melanocytes, especially in the cluster associated with pigmented
cells, and identified biological processes shared between these cell types. By performing
phylogenetic tree reconstruction using transcription factors shared between rodents and
tunicates, they showed that RPE and *Ciona* pigmented cells are neighbors, with melanocytes
as the closest outgroup.

Finally, they implemented a phylostratigraphy approach to explore the evolutionary age of
genes and programs specific to RPE and melanocytes, determining the genes or programs
that emerged after the putative split of pigmented photoreceptors into RPE and melanocyte
lineages. Based on the set of genes identified, the authors claim that functions specific to
melanocytes and RPE were acquired after that split. Overall, the authors suggest that since

RPE and melanocytes share common programs and function-specific genes emerged later,
RPE and melanocytes are likely to share the same origin.

Although the authors made efforts to investigate multiple datasets, the manuscript seems to
lack strong data supporting their hypothesis. The proposed strategy is interesting, the data
presented in the manuscript fall a bit short to be really convincing that melanocytes and RPE
share a common cell type of origin and that evolutionary convergence is to be excluded.

- We sincerely thank the reviewer for their stimulating suggestions and thoughtful
directions, which have significantly contributed to this study. As is often the case in the
field of evolutionary developmental biology, definitive proof is rarely attainable, and
hypotheses can seldom be confirmed with absolute certainty. Instead, the aim is to
gather converging lines of evidence that increase the likelihood of one evolutionary
scenario over another - a limitation that reflects the current state of the field.

**In response to the reviewer's suggestions, we made a concerted effort** to further
explore the potential homology between retinal pigment epithelium cells and
melanocytes. We have included additional analyses of **compared chromatin states**
and **tested our parsimony-based comparative framework** using datasets with
evolutionary ground truths (neural retina, see doi:10.1038/nrg.2016.127). Please see
**new Supplementary Figures 2,4,5,6.**

**Importantly, we incorporated a new paragraph** that explicitly acknowledges the
limitations of our findings:

*“At last, we would like to address the limitations of this study. In evolutionary biology*
*research, evidence is most often suggestive rather than direct and conclusive. The*
*complexity of evolutionary processes means that direct evidence, such as fossil*
*records linking specific molecular traits to ancestral species, is often missing or*
*incomplete. As a result, researchers rely on indirect evidence, such as similarities in*
*gene expression profiles, regulatory elements, or cell types between distantly related*
*species. This study is no exception. Furthermore, comparative evolutionary single-cell*
*biology is still in its developmental stages, and key datasets remain unavailable - for*
*example, single-cell transcriptomics covering amphioxus photosensory organs and*
*cyclostome pigment systems. Therefore, this study should be regarded as a foundation*
*for future work rather than definitive proof of homology between melanocytes and RPE.*
*Further progress will require a robust evolutionary framework, including genome-wide*
*regulatory landscape analyses, which are not yet fully achievable.*

*We encourage readers to interpret our findings as a compelling line of evidence*
*suggesting a connection between the evolutionary origin of the neural crest and*
*melanin-containing photosensory cells - a hypothesis that merits further testing as the*
*field of cross-species single-cell analysis continues to evolve and mature.”*

While our results do not provide conclusive 100% evidence of homology (as there is
always a slim chance for unlikely set of unrelated co-options), they do suggest a high
degree of plausibility, which we believe merits further investigation, particularly in the
context of the unresolved questions surrounding neural crest cell origin. We hope these
revisions and clarifications address the reviewer's concerns and reflect the
improvements made in response to their valuable feedback.

**Major points**

1. The authors showed that RPE and melanocytes have common transcriptional programs.
However, this is essentially based on melanin production ribosomes biogenesis and metabolic
processes. This evidence seems insufficient to favor the common ancestry hypothesis over
the convergent evolution one. To support their claim, the authors need to compare – if such
exist – cell types known to have a common ancestry as a reference point.

- We agree and we compared cell types sharing the known common ancestry using our
parsimony-based tool and, as an alternative, hierarchical clustering. Please see **new**
**Supplementary Figure 3**, where we added the evolutionary-connected cell types of
neuroretina.

The following text in Results section describes this new analysis:

*“To further assess such constructed tree's ability to capture known evolutionary*
*relationships, we extended the dataset to include a broader range of cell types, such*
*as rod and cone photoreceptors, retinal neuroglia, and immune cells from Li et al.,*
*2024 (Li et al., 2024) (Supplementary Fig. 6a-b). After integrating retinal dataset by*
*Li et al. 2024 with the anterior eye chamber dataset by Lee et al. 2022 (Lee et al.,*
*2022) from Fig.1 (Supplementary Fig. 6c), we successfully recapitulated previously*
*proposed phylogenetic relationships, including the close pairing of rods and cones*
*(Arendt et al., 2016), the grouping of astrocytes with Müller glia (Pfeiffer et al., 2020),*
*and the segregation of various immune cell classes, providing additional support for*
*the robustness of the method (Supplementary Fig. 6d).”*

2. The study focuses on the eye and compares RPE and melanocytes from the same organ.
The fact that these cells are localized in the same region could also explain why they share
transcriptional programs, as they are exposed to the same surrounding signals and
environment. The authors need to add data from skin melanocytes in their analysis to prove
this similarity in transcriptional programs is really reflecting a putative common origin or if on
the contrary it reflects an evolutionary convergence linked to the cell being positioned in the
same organ. Identifying common programs among these three cell types would support the
claim better than just comparing eye cell types.

- We thank the reviewer for this suggestion, and we also think that such comparison is
a smart thing to do. Indeed, the location can dictate some environment-related
similarities, and we would like to avoid that. We performed the requested analysis, and
the Results can be found in the **new Supplementary Figure 2** and in the main text:

*“We next sought to determine whether the observed transcriptional similarity between*
*retinal RPE cells and ocular melanocytes reflects a shared evolutionary origin, or*
*alternatively, results from convergent transcriptional programs driven by a common*
*tissue environment - specifically, their co-localization within the eye and exposure to*
*similar signaling niches. To address this question more rigorously, we extended our*
*comparison beyond ocular tissues by incorporating skin-derived melanocytes into the*
*analysis. If RPE cells cluster more closely with melanocytes from distant anatomical*
*sites, such as the skin, this would support the notion of deep lineage conservation,*
*rather than local convergence due to tissue-specific niche signals.*

*To systematically compare RPE cells with skin melanocytes in the context of diverse*
*cellular lineages, we merged a dataset from the anterior eye chamber containing RPE*
*cells (GSM5560840, GEO / (Lee et al., 2022)) with four distinct clusters of skin-derived*
*melanocytes obtained from published sources. These included: melanocytes from*
*back skin at postnatal day 0 (GSE131498, GEO / (W. Ge et al., 2020)), postnatal day*
*4 back skin melanocytes (GSE181390, GEO / (Lee et al., 2024)), activated melanocyte*
*stem cells from anagen II stage hair follicles in five-week-old mice (GSE203051, GEO*
*/ (Sun et al., 2023)), and quiescent melanocyte stem cells profiled via Smart-seq2*
*(GSE147298 / (Infarinato et al., 2020)). Each melanocyte cluster from these studies*
*was integrated into the analysis independently (Supplementary Fig. 2).*
*Hierarchical clustering of the merged data consistently positioned RPE cells in close*
*proximity to both ocular and skin-derived melanocytes, relative to all other cell types.*
*After constructing a dendrogram based on hierarchical clustering of global*
*transcriptional profiles (considering all genes), we observed that RPE cells,*
*melanocytes from both ocular and skin origins, and Schwann cells consistently*
*clustered in close proximity (Supplementary Fig. 2a-c, e-g, i-k, m-o).”*

Notably, parsimony-based analysis performed with CASSIOPEIA, based on
transcription factor profiles, consistently grouped RPE cells with skin-derived
melanocytes across all datasets. Interestingly, in these analyses, skin-derived
melanocytes demonstrated a greater transcriptional similarity to RPE cells than did
eye-derived melanocytes.

3. The authors based their analyses solely on gene expression datasets. Changes in
sequences of distal regulatory elements has been showed to be a major driver of evolution as
it can happen much more frequently than mutation in transcription factor binding or changes
in gene function. In the present study, chromatin accessibility and regulatory elements have
not been studied. Comparing and analyzing such datasets could enrich the study, as revealing
common processes at this level would provide additional evidence for a shared origin.

- We thank the reviewer for this insightful suggestion, which we followed to perform a
series of comparative analyses of chromatin states in RPE and melanocytes. To do
so, we leveraged the recently published dataset by Mullin et al. (2023), which provides
high-resolution 10x Multiome data (simultaneous ATAC-seq and single-nucleus RNA-
seq) from the mouse eye. This dataset offers chromatin accessibility landscapes for
ocular melanocytes, RPE cells, and numerous other neuroretinal and anterior eye
chamber cell types. A key advantage of this dataset is the high cell type resolution
afforded by single-nucleus transcriptomics sequencing, which allows for more accurate
cell type identification compared to ATAC-seq-only data. The **new analysis** can be
found in **Supplementary Figures 4 and 5**.

For instance, here is the part of the text of Results describing the comparison of
chromatin accessibility landscapes:

*“Changes in the sequences of distal regulatory elements have been recognized as*
*major drivers of evolutionary diversification across species (Mack and Nachman,*
*2017). Thus, we explored the regulatory landscape of RPE and melanocytes in a*

*comparative context, focusing on chromatin accessibility and non-coding regulatory*
*elements that may contribute to their transcriptional similarity and evolutionary*
*relationships. To achieve this, we utilized the recently published high-resolution 10x*
*Multiome dataset from Mullin et al. 2023 (Mullin et al., 2023), which simultaneously*
*profiles chromatin accessibility (ATAC-seq) and gene expression (single-nucleus*
*RNA-seq) from the mouse eye (Supplementary Fig. 4). This dataset offers*
*comprehensive coverage of ocular melanocytes, RPE cells, and a broad range of*
*neuroretinal and anterior eye chamber cell types. A key advantage of this multiomic*
*approach is the ability to confidently assign cell identities based on transcriptomic*
*signatures while analyzing chromatin accessibility in the same nuclei, thus overcoming*
*limitations inherent to ATAC-seq-only data, such as low cell-type resolution and*
*susceptibility to misclassification.*
*Our initial dimensionality reduction analysis, based solely on ATAC-seq embeddings,*
*revealed substantial misattribution between RPE and melanocyte clusters, with cells*
*frequently grouped into one another's domains (Supplementary Fig. 4a-b). However,*
*by cross-referencing transcriptomic profiles from the same nuclei, we were able to*
*resolve true cell identities with higher confidence. These observations prompted the*
*hypothesis that RPE cells and melanocytes share a significant fraction of accessible*
*chromatin regions, particularly in non-promoter, non-coding elements, which may*
*underlie their apparent proximity in ATAC-based embeddings. To formally assess this,*
*we constructed a dendrogram using a maximum distance tree built from chromatin*
*accessibility data restricted to non-promoter intergenic regions, because promoter-*
*related regions will reflect overall expression level of corresponding genes. This*
*analysis revealed that RPE cells, melanocytes, and Schwann cells clustered on the*
*dendrogram in close proximity (Supplementary Fig. 4c). The focused investigation of*
*open chromatin regions in the upstream and downstream regions of Sox10 – a*
*transcription factor with a key role in both melanocytes and RPE cells – revealed mostly*
*common and some distinctly open chromatin regions linked to promoter accessibility*
*in peaks-linked-to-genes analysis (Supplementary Fig. 4d)...”*

Overall, our analysis revealed that RPE cells, melanocytes, and Schwann cells
clustered closely together based on chromatin accessibility. The proximity of Schwann
cells is biologically expected, given that melanocytes are developmentally derived from
the Schwann cell lineage (see Adameyko et al., Cell 2009; Adameyko et al.,
Development 2012), and these cell types retain transcriptional similarity reflective of
their shared developmental origin. This outcome provides confidence in the robustness
of our chromatin-based analysis. To further assess this result, we applied an
alternative method based on Euclidean distance using the same subset of accessible
chromatin regions. In this case, RPE cells formed a highly distant outgroup, yielding a
dendrogram with limited interpretability in regard to RPE. This discrepancy likely
reflects the inherent noise in ATAC-seq data, particularly when used in isolation. In
contrast, our transcriptomic comparisons using the CASSIOPEIA tool yielded more
stable and interpretable clustering with both ocular and skin melanocytes.

Also, to overcome this, we further tested CASSIOPEIA's performance in this revision
using an extended dataset containing different classes of evolutionary related
photoreceptors (rods and cones) and neuroglial retinal cells together with RPE and

melanocytes (see **new Supplementary Figure 6**). This analysis captured previously
suggested evolutionary relationships between rods and cones (Arendt et al., 2016),
different subtypes of immune cells, as well as astrocytes and Müller glia.
We also performed gene ontology analysis to explore the biological relevance of these
shared open chromatin regions. The enriched pathways included pigmentation-related
processes, cytoskeletal regulation, and immune-related signaling - results that are
biologically plausible and in line with prior observations. These findings are further
supported by phylostratigraphy analysis, which indicates that evolutionarily recent
genes involved in immune pathways have been independently recruited into both RPE
and melanocyte cell types. These new results are presented in **Supplementary**
**Figures 4 and 5**.

4. The authors used the *Ciona* larva as an outgroup. They highlighted a set of markers for
melanocytes and RPE in *Ciona* larva, particularly in the pigmented photoreceptor cluster.
Images showing the expression of these markers in the larva would be interesting to estimate
the specificity of these markers within the organism and identify other cell types expressing
them. This is essential since there are many examples in the literature showing that ancestral
genes in amphioxus or lamprey are expressed during development but either at different
timing or location or have even have different functions.

- As advised by the reviewer, we performed HCR hybridizations with RNA probes for
several key genes that we use to outline photosensory and pigmented cell types in
*Ciona*: *Tyr* (pigmentation), *Otx* (anterior neural tube marker), *Rpe65* (RPE marker),
*Rrh* (photoreception) and ependymal neuroprogenitors marker *Hh2*. These markers
showed an overlapping expression in the sensory vesicle, more specifically in
pigmented photosensory cells (except for *Hh2* as expected). These results support the
logic of our study and help to homologize pigmented ocellus of *Ciona* with vertebrate
RPE and melanocytes (please find these new results in **revised Figure 5e**).

5. The manuscript would greatly benefit from a detailed description (rather than an overview)
of all figures, as well as thorough proofreading of the abstract, text, and figure legends. Several
passages are extremely deterministic with conclusion drawn without evidences. Finally,
several paragraphs needs to be rewritten in a more scientific style. For example, a sentence
like “Needless to say that » are really not welcome in a scientific manuscript.

- We have improved the descriptions of the results and figures, and have revised the
main text to enhance overall clarity. We hope the reviewer finds these improvements
evident in the revised manuscript. Importantly, we have clearly stated that while our
findings support the proposed homology between RPE cells and melanocytes, they do
not constitute definitive proof. We have emphasized this point explicitly in the
concluding paragraph of the Discussion, and we trust the reviewer will appreciate the
added clarity. At the same time, the plausibility of this homology is reinforced by the
shared presence of photosensory pathways - beginning with opsins and extending
through components of the phototransduction machinery. This conserved “visual
function”, along with a suite of other shared metabolic and biochemical modules that
appear unrelated to pigmentation or vision, strengthens the case for a common origin,
even when pigmentation is not considered the central unifying trait. Following the

reviewer's valuable suggestions, we also conducted additional analyses and
uncovered further evidence of similarity in the chromatin accessibility landscapes of
RPE cells and melanocytes. This new finding adds another layer of support to the
proposed idea of homology.

Overall, we are grateful for the reviewer's insightful feedback, which played a pivotal
role in refining and elevating the quality of this study. Thank you once again for your
constructive and thoughtful comments.

Response to the referees (the final minor revision)

Reviewer #1 (Remarks to the Author):

The authors have significantly improved the evidence for deep homology of RPEs and melanocytes. The new results presented in Fig. 5 are a nice addition to the revised manuscript. My only lingering concern is the length of the Discussion. It is currently 5 pages long and could easily be trimmed to 3 pages by eliminating repetitions with the Intro and Results section.

- First of all, we cordially thank the reviewer for helping us to improve this study. We generally agree with the reviewer, and we tried to shorten the Discussion by removing everything repetitive. However, during the previous round of revision, we had to add new reasoning for Reviewers 2 and 3 (as requested), and that took up some volume again, approximately 1 additional page. Given this uneasy situation, we found the most reasonable compromise and now have only 4 pages of the Discussion (1 page of the text was removed).

Reviewer #2 (Remarks to the Author):

In their revised manuscript Fatieieva et al. carefully addressed my comments and those of the other reviewers. They added substantial new data analyses and completely rewrote the manuscript, which is now significantly improved.

- We thank the reviewer for this optimistic opinion, and we feel grateful for all comments and helpful instructions.

I have only a few minor comments remaining:

- 123-126: I think homology is about identity rather than similarity and would thus prefer to word this differently (although I accept that the authors may disagree), such as: "Homology refers to the identity of structures in different species due to common ancestry, which may not be immediately apparent in distantly related species, where structures may have become dissimilar due to evolutionary convergence" and provide a different reference (e.g . Wagner, G. P. 2014. Homology, Genes, and Evolutionary Innovation. Princeton: Princeton University Press).

- We agree, and we inserted this exact phrase as advised, and also added the reference to prof. Wagner's famous book, which we have on our shelf.

- 470-473: This sentence confuses rather than clarifies what the authors propose. In the next paragraph they state that skeletogenic fates may have been coopted secondarily into the NC lineage. So it is

confusing that they write here “and beyond, as seen in the case of mesoderm-like ectomesenchyme”. I also find it misleading to talk about “functionally analogous” cell types here, since their central argument seems to be about homologous cell types. Thus I suggest to simplify and reword the sentence as follows: “This idea is supported by the observation that the neural crest frequently generates cell types that are identical to those produced by other progenitor populations within the CNS.”

- Yes, this is corrected now exactly as suggested by the Reviewer.

- 507: delete “in lineage”

- Deleted

- 517: add “we” before performed”

- Added

- There are still a few typos in the figures or figure legends (e.g. for Cassiopeia in Fig. 3c, Suppl. Figs 2 and 6); for pericyte in Fig. 1a, for melanocyte in Fig. 7a).

- All corrected

- abbreviation “ov” in Fig. 5d needs to be explained in legend

- We thank the reviewer for spotting this. OV shall be SV (sensory vesicle). This is now corrected OV for SV in the figure.

- the lower part of Suppl. Fig. 4d is not explained in legend

- We explained the lower part of Suppl. Fig. 4d explicitly in the revised figure legend.

Reviewer #3 (Remarks to the Author):

The authors addressed all my remarks in the revised version of this manuscript which now is much clearer and convincing. It is now ready for publication. Some minor comments below that need to be corrected in the text:

- We thank our reviewer for all constructive advice and an interesting discussion.

1. Line 160: “Although these genes might be stronger enriched “

- Rewritten and corrected

2. Line 162 – 163: “Gene ontology enrichment and STRING analysis of these common melanocyte (MEL) + RPE genes revealed that”

- This phrase is rewritten and corrected

3. Line 516-517: The term “pools of DNA Controllers » is very unclear. Authors should use transcription factor or DNA binding proteins instead

- Corrected for DNA-binding proteins

4. Line 523: When using the term “transcriptional controllers” are the authors referring to mechanisms regulating the rate of transcription or to DNA binding proteins? The authors need to clarify this sentence

- Corrected for DNA-binding proteins:

“The cladistics approach to DNA-binding proteins (most of which are transcriptional controllers) simplifies the comparisons of cell types with complex gene expression profiles...”

5. Line 533: The authors need to replace the term “regulatory controllers” by something more generally used by the community

- Corrected for DNA-binding proteins

6. Fig.2c: $-\log_{10}(\text{FDR})$ scale value is missing

- We thank the reviewer for spotting this. The scale is added to the Figure 2c.

7. Fig. 5d: It is indicated Melano but authors should write Melanocytes

- Corrected for Melanocytes